# Hyperspectral Remote Sensing Detection of Marine Oil Spills Using an Adaptive Long-Term Moment Estimation Optimizer

**Zongchen Jiang [1,2], Jie Zhang [1,2,3,4,*], Yi Ma [2,3,4] and Xingpeng Mao [1]**

1    School of Electronics and Information Engineering, Harbin Institute of Technology, Harbin 150001, China; jzc@fio.org.cn (Z.J.); mxp@hit.edu.cn (X.M.)
2    Remote Sensing Department, The First Institute of Oceanography, Ministry of Natural Resources, Qingdao 266061, China; mayimail@fio.org.cn
3    Ocean Telemetry Innovation Technology Center, The First Institute of Oceanography, Ministry of Natural Resources, Qingdao 266061, China
4    College of Oceanography and Space Informatics, China University of Petroleum (East China), Qingdao 266580, China
*    Correspondence: zhangjie@fio.org.cn

**Abstract:** Marine oil spills can damage marine ecosystems, economic development, and human health. It is important to accurately identify the type of oil spills and detect the thickness of oil films on the sea surface to obtain the amount of oil spill for on-site emergency responses and scientific decision-making. Optical remote sensing is an important method for marine oil-spill detection and identification. In this study, hyperspectral images of five types of oil spills were obtained using unmanned aerial vehicles (UAV). To address the poor spectral separability between different types of light oils and weak spectral differences in heavy oils with different thicknesses, we propose the adaptive long-term moment estimation (ALTME) optimizer, which cumulatively learns the spectral characteristics and then builds a marine oil-spill detection model based on a one-dimensional convolutional neural network. The results of the detection experiment show that the ALTME optimizer can store in memory multiple batches of long-term oil-spill spectral information, accurately identify the type of oil spills, and detect different thicknesses of oil films. The overall detection accuracy is larger than 98.09%, and the Kappa coefficient is larger than 0.970. The $F_1$-score for the recognition of light-oil types is larger than 0.971, and the $F_1$-score for detecting films of heavy oils with different film thicknesses is larger than 0.980. The proposed optimizer also performs well on a public hyperspectral dataset. We further carried out a feasibility study on oil-spill detection using UAV thermal infrared remote sensing technology, and the results show its potential for oil-spill detection in strong sunlight.

**Keywords:** hyperspectral remote sensing; marine oil spill; oil film thickness detection; oil spill type identification; deep learning

## 1. Introduction

With the rapid development of the global marine transportation and offshore oil extraction industries, marine oil spills frequently occur, which seriously affects the sustainable development of the marine ecological environment and resources. The accurate identification and analysis of the type of marine oil spills are helpful for determining the responsibility for accidents and are extremely important for on-site emergency responses and the rapid and effective treatment of sea surface pollution. Obtaining an accurate value for the oil film thickness and then estimating the amount of oil spill is an important basis for accountability in pollution compensation, which plays an important role in scientific decision-making and determining the severity of the oil-spill accident [1,2]. Oil-spill type identification and oil-film thickness detection via remote sensing are popular topics at the frontier of current research on oil-spill optical remote sensing, which remains susceptible

to cloud and fog conditions [3–6]. Unmanned aerial vehicle (UAV) optical remote sensing technology has been used for the disposal of oil-spill accidents in recent years because it can quickly and effectively acquire remote sensing data of marine oil spills [7,8].

Lammoglia used laboratory measurements to obtain the spectral data of different oil types, developed a series of feature extraction and identification methods for different oil types based on principal component analysis, and established an oil type spectral library. Experiments have shown that the spectral absorption characteristics of crude and fuel oils are obvious, but the effective identification of different types of light oils has not yet been achieved [9–11]. The current standard for crude oil film thickness evaluation is the Bonn protocol, which is recognized by the International Maritime Organization. The main problem with the protocol is that the oil film identification is strongly affected by subjective and environmental factors, and it is impossible to distinguish between thick oil films larger than 100 μm [12,13]. Lu's experimental results show that within a certain range, the absolute thickness of the oil film has a positive linear relationship with the level of brightness temperature, but the upper limit of this detection method is 400 μm [14]. Lu further measured oil emulsion spectra in a laboratory environment to obtain non-imaging ASD hyperspectral data of crude oil emulsions that can fully represent the backscattering characteristics as well as C-H and O-H absorption characteristics [15]. A hyperspectral remote sensing identification method of oil emulsion based on a decision tree was proposed, and good experimental results were obtained [16].

An oil spill is a weak target in optical remote sensing research, and it is difficult to distinguish the spectral curves of different types of light oils. Traditional remote sensing modeling methods based on characteristic bands have made positive progress with heavy oils, but the research on identifying light oil types has been mostly carried out in a laboratory environment, and the performance of the proposed methods is not ideal [17–23]. In addition, the oil film of heavy oil has a strong absorption effect on sunlight, so the remote sensing reflectance is lower than light oils, and the spectral separability of oil films with different thicknesses is poor. Therefore, it is difficult to detect the thickness of thick oil film based on optical remote sensing. At present, the research in oil film thickness is mostly based on the inversion of thin or relative thickness based on traditional remote sensing modeling methods, and the research on the detection of the absolute thickness of thick oil film has not been adequately investigated [24–27].

Hyperspectral remote sensing has the characteristics of high spectral resolution and a wide spectral response range. In contrast to traditional multi-spectral remote sensing technology, it can obtain rich oil-spill spectral characteristic information [28–30]. Moreover, deep learning has developed rapidly in recent years because of its powerful ability to extract features from high-dimensional data [31–37]. Deep networks and multi-level features fusion method for deep learning have been applied to hyperspectral image classification, and research progress has been made [38–40]. In this paper, the combination of hyperspectral remote sensing technology and deep learning is conducive to extracting the spectral characteristic information and then accurately identifying the type of oil spills and detecting thick oil films with different thicknesses.

At present, marine oil-spill detection based on remote sensing mostly lacks on-site measured data to carry out accurate accuracy verification of detection methods [41,42]. In this study, an outdoor experimental setup was built to simulate a real marine oil-spill environment. By adding experimental oil qualitatively and quantitatively to the experimental scene, the accuracy of the detection results of the oil spill experiment will be evaluated. Compared to non-imaging oil spill hyperspectral studies, we collected airborne hyperspectral remote sensing images of five types of oil spills (crude oil, fuel oil, gasoline, diesel, and palm oil), crude oil, and fuel oil with different thicknesses at three different times.

The traditional adaptive moment estimation (Adam) optimizer, which is widely used in various deep learning tasks, uses the mechanism of an exponential moving average to update the second momentum term and then update the model weight. When applied

to the hyperspectral oil-spill images in this experiment, the oil-spill spectral characteristic information would be lost, which would affect the oil-spill detection performance [43–45]. We combine UAV hyperspectral remote sensing technology with deep learning and propose an adaptive long-term moment estimation (ALTME) optimizer which can meet the characteristics of hyperspectral imagery and adapt to different phases of oil-spill scenes. To realize the cumulative learning function of the optimizer for multi-batch and long-term historical spectral characteristic information, we integrate the proposed optimizer with a one-dimensional convolutional neural network (1D-CNN); we consequently extracted the oil-spill spectral characteristic information obtained by the hyperspectral remote sensing data using 1D convolution. The aim of the marine oil-spill 1D-CNN detection model is to improve the detection performance of oil-spill types and thick oil films, and it is expected to be applied in future research on the detection of marine oil spills.

## 2. Data and Method

### 2.1. Data

The experiment lasted for four days, from 19 to 22 September 2020. We conducted the experiment in the large-scale land-based experimental pool of Qingdao Scientific Research Base, China, which is 50 m long, 40 m wide, and 2 m deep. There is no shade during the day, and the lighting conditions are good. About 20 m to the west of the pool is the Yellow Sea shore, which facilitates the extraction of seawater required for the experiment. An aerial photograph of the experimental land-based pool is shown in Figure 1.

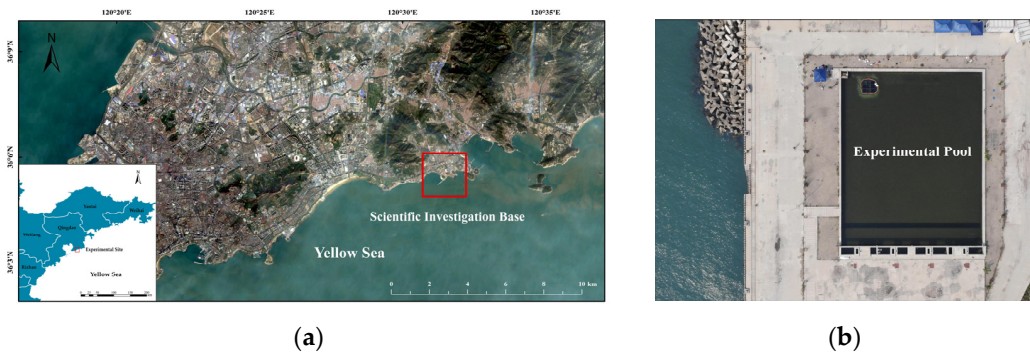

(**a**)　　　　　　　　　　　　　　　　　　　　　　　(**b**)

**Figure 1.** (**a**) Location of the experimental site and (**b**) aerial view of the experimental pool.

In this experiment, a Cubert-S185 hyperspectral sensor (Cubert, Berlin, Germany) was used to obtain the hyperspectral images of the oil spills on the sea surface. The radiometric calibration plate is AZ-WS20, and its manufacturer is Sphereoptics from the Washington, DC, USA. DJI-M600PRO UAV is equipped with an A3Pro flight control system and Zenmuse gimbal, which has good stability and balance ability. The sensor was installed on a DJI-M600PRO UAV, as shown in Figure 2.

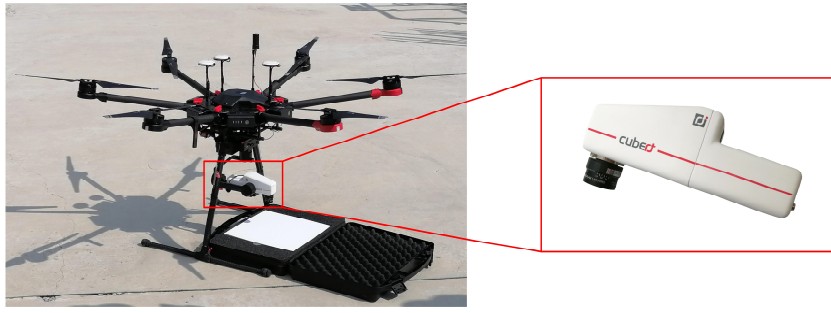

**Figure 2.** DJI-M600PRO UAV and Cubert-S185 hyperspectral sensor.

The Cubert-S185 hyperspectral sensor uses a frame imaging mode; it has a spectral range of 450–950 nm, a spectral resolution of 4 nm, and 125 bands. The specifications of the Cubert-S185 sensor and M600PRO UAV are listed in Table 1. In addition, using the 12 megapixels CMOS optical imaging system carried by DJI-Mavic2 UAV, we collected high-resolution optical images of the oil spills to assist the ground-truth labeling of the hyperspectral images so as to carry out accuracy verification of the detection model proposed in this paper.

**Table 1.** Specifications of Cubert-S185 Hyperspectral Sensor and DJI-M600PRO UAV.

| Parameter | | Index |
| --- | --- | --- |
| Cubert-S185 | Spectral range (nm) | 450~950 |
| | Spectral resolution (nm) | 4 |
| | Number of bands | 125 |
| | IFOV (°) | 23 |
| | Imaging method | Frame imaging |
| | Imaging size (pixel) | 1000 × 1000 |
| M600PRO | Maximum load (kg) | 6 |
| | Flight duration (min) | 16 |
| | Maximum wind resistance level (m/s) | 8 |
| | Maximum ascent/descend speed (m/s) | 5/3 |
| | Maximum flight altitude (m) | 4500 |
| | Maximum horizontal speed (km/h) | 65 |

While ensuring the safety of the experiment, our aim was to simulate a real marine oil-spill accident scene as closely as possible. Because the experiment involves the identification of oil-spill types and the detection of oil films with different thicknesses, to carry out qualitative remote sensing analysis more ideally, we built an experimental enclosure using a PVC (Polyvinyl Chloride) board. The PVC enclosure was sealed and reinforced with high-strength glass glue and hinges to prevent the oil films from spreading irregularly on the water surface, and an image of its construction is shown in Figure 3. The experimental enclosure has a height of 1.2 m and consists of nine small enclosures. The size of each small enclosure was 1 × 1 m, which we labeled as groups 1–9 in turn. Because the experimental site is close to the Yellow Sea and Laoshan Bathing Beach, to reduce the risk of oil spills, we set up an oil containment boom around the experimental device to prevent oil films leakage, as shown in Figure 4.

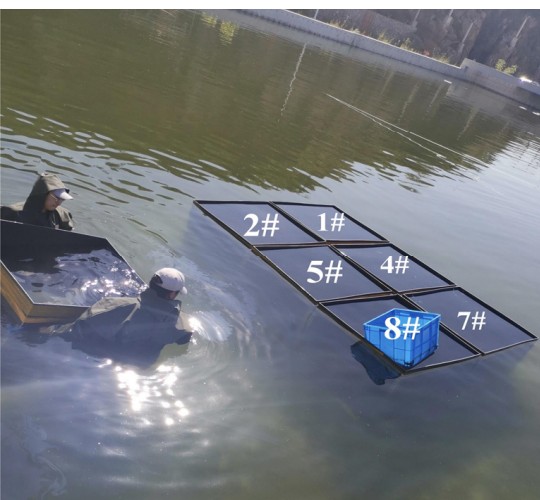

**Figure 3.** Installation of the PVC device.

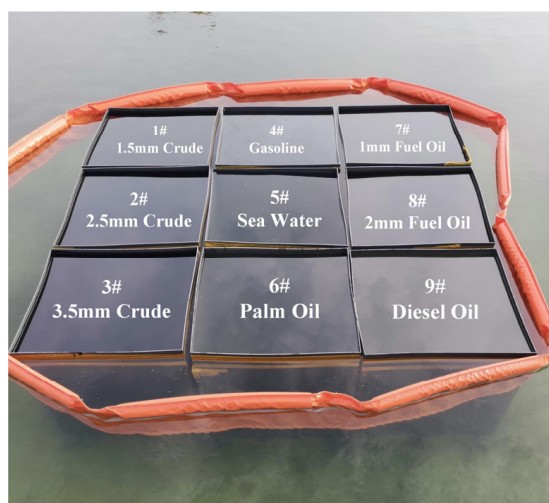

**Figure 4.** Oil spill experiment group labels.

The oils used in this experiment are shown in Figure 5. From left to right are crude oil from the Dongying Shengli Oilfield, fuel oil, edible palm oil, diesel (#0), and gasoline (#95). The heavy-oil experimental groups included crude oil and fuel oil, and the light-oil experimental groups included palm oil, diesel, and gasoline.

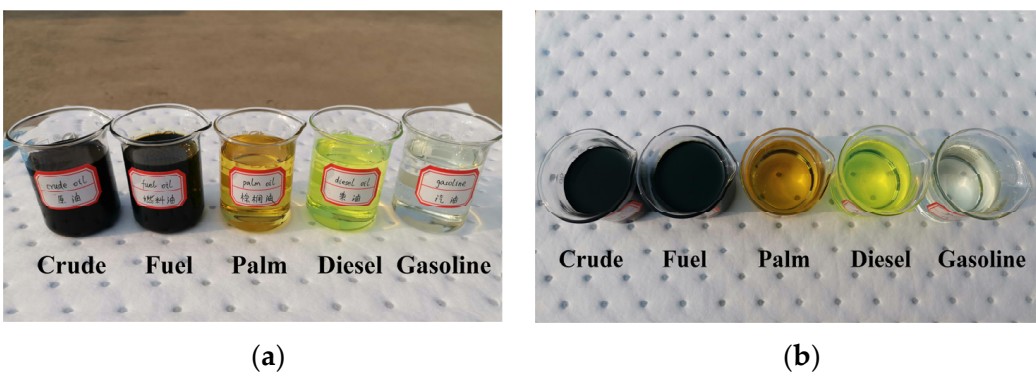

(a)                                                    (b)

**Figure 5.** (**a**) Front view of the experimental oils and (**b**) top view of the experimental oils.

The Bonn protocol distinguishes in detail the thin oil films of crude oils smaller than 0.1 mm (the protocol has seven levels), whereas the distinction for black oil slicks of 0.5–4 mm is rough (only two levels are used). Our aim was to investigate the hyperspectral remote sensing detection of thick heavy-oil films. Hence, we set the absolute thicknesses of the crude oil films in the experiment to be approximately 1.5 mm, 2.5 mm, and 3.5 mm, with thickness intervals of 1 mm. The absolute thickness of the fuel oil films was set to approximately 1 mm and 2 mm with a thickness interval of 1 mm. In this study, we regarded oil films with different thicknesses as different thickness grades to explore the capabilities of the airborne hyperspectral remote sensing technology to qualitatively detect thick heavy-oil films.

In this study, under the same experimental environment, the density of the crude and fuel oils was calculated to be 0.81 kg/L and 0.85 kg/L, respectively. A pallet balance (with an accuracy of 0.001 kg) was used to calculate the quality of the experimental oil by measuring the difference before and after it was poured into the experimental enclosure, and then the volume was obtained based on the density of the oil. The weighing process of crude oil and fuel oil is shown in Figure 6.

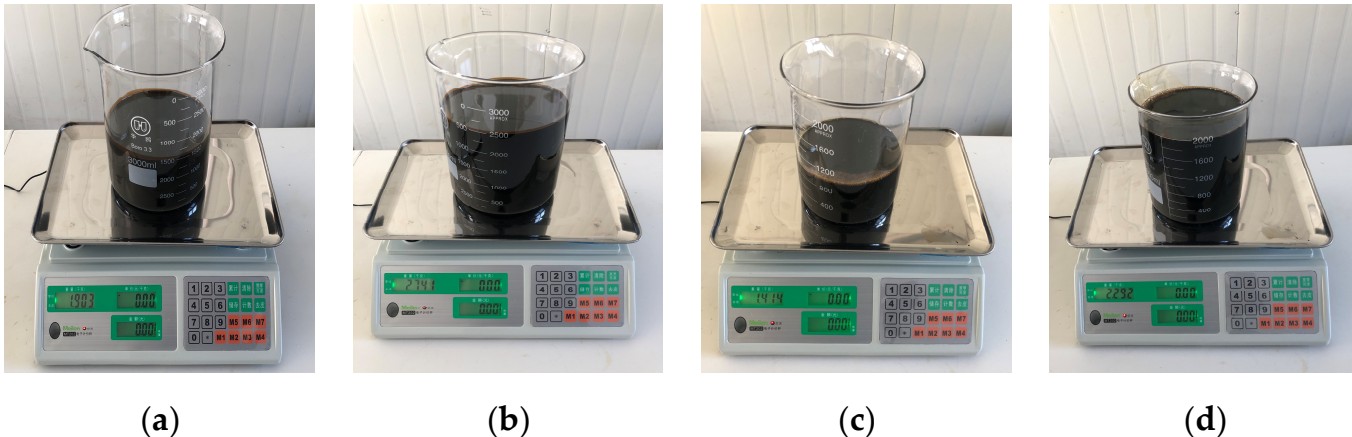

**Figure 6.** Measuring the quality of the experimental oils. (**a**,**b**) are the process of crude oil measurement. (**c**,**d**) are the process of fuel oil measurement.

The size of each small enclosure ($S_E$) in the experiment is known to be 1 m². We used large-capacity beakers (2 L and 3 L) to control the volume of oil poured into the experimental enclosure. We then calculated the absolute thickness of the oil films of the heavy-oil experimental groups using

$$h_{Oil} = \frac{M_1 - M_2}{\rho_{Oil} \cdot S_E} \tag{1}$$

where $M_1$ is the mass of the beaker before adding the experimental oil, $M_2$ is the mass of the beaker after adding the oil, $\rho_{Oil}$ is the density of the oil, and $h_{Oil}$ is the absolute thickness of the oil film.

With the exception of group 5, which was seawater without any experimental oil, we poured crude oil into groups 1–3, #95 gasoline into group 4, palm oil into group 6, fuel oil into groups 7–8, and #0 diesel into group 9 in the experimental enclosure in sequence to obtain oil-spill observation scenes of the different types of light oils and heavy-oil films with different thicknesses. The measured data of the heavy-oil film experimental groups is given in Table 2.

**Table 2.** Data of the heavy-oil film experiment groups.

| Group | $M_1$ (kg) | $M_2$ (kg) | $M$ (kg) | $\rho_{Oil}$ (kg/L) | $V$ (L) | $h_{Oil}$ (mm) |
|---|---|---|---|---|---|---|
| 1#-1.5 mm Crude | 1.90 | 0.66 | 1.24 | 0.81 | 1.53 | 1.53 |
| 2#-2.5 mm Crude | 2.74 | 0.65 | 2.09 | 0.81 | 2.58 | 2.58 |
| 3#-3.5 mm Crude | 4.10 | 1.21 | 2.89 | 0.81 | 3.57 | 3.57 |
| 7#-2 mm Fuel Oil | 1.41 | 0.58 | 0.83 | 0.85 | 0.98 | 0.98 |
| 8#-3 mm Fuel Oil | 2.30 | 0.58 | 1.72 | 0.85 | 2.06 | 2.06 |

After uniform diffusion of the oil films, using the calibrated Cubert-S185 hyperspectral sensor carried by the DJI-M600PRO UAV, we obtained hyperspectral images of the oil films in the enclosure at 11:30 h (T1), 14:00 h (T2), and 16:30 h (T3). In this experiment, we took the flying method of avoiding the sun glint to avoid the influence of the sun glint. The time intervals between T1, T2, and T3 are all 2.5 h, and the experimental process of data acquisition is shown in Figure 7.

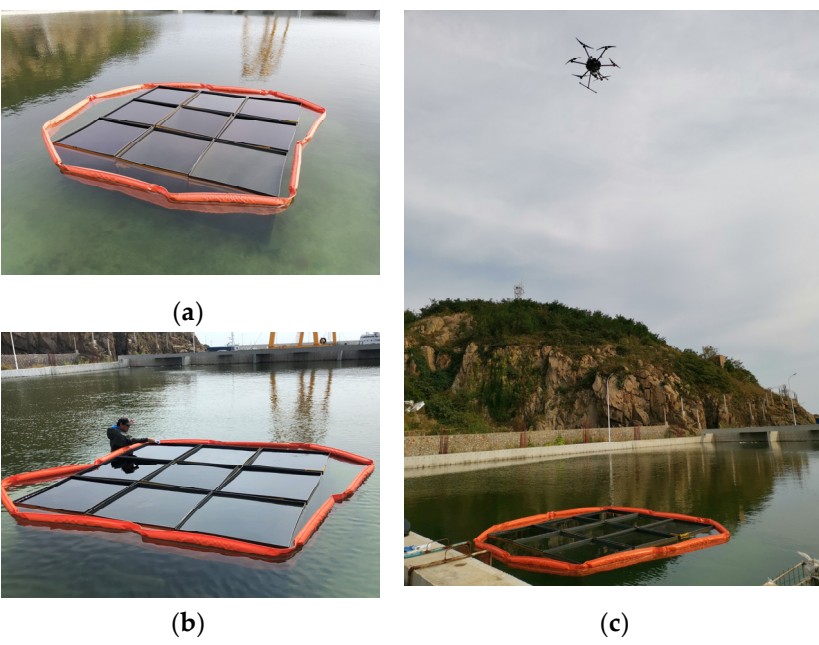

**Figure 7.** (**a–c**) are on-site photos of the experiment.

Environmental data such as wind speed and temperature were obtained with a Kestrel-3500 meteorological instrument (Table 3). During the UAV hyperspectral remote sensing data collection, the weather was clear, and there was no cloud or fog obstruction. From T1 to T3, the wind speed gradually increased, but the overall wind speed and wind speeds in all directions were below 2 m/s, which is classed as "light air" according to the Beaufort Scale, and the sea was relatively calm.

**Table 3.** Environmental data of the experiment.

| Moment | Wind Speed (m/s) | Cross Wind (m/s) | Upwind (m/s) | Temp (°C) | Solar Elevation Angle (°) | Solar Azimuth Angle (°) | Weather |
|---|---|---|---|---|---|---|---|
| 11:30 | 0.80 | 0.70 | −0.30 | 26.80 | 53.82 | 12.77 | Sunny |
| 13:50 | 1.50 | 0.80 | 1.30 | 26.90 | 45.92 | −41.59 | Sunny |
| 16:21 | 1.80 | 1.00 | 1.50 | 24.50 | 19.84 | −74.89 | Sunny |

In this study, using Python and the Cubert-Touch platform, radiometric calibration, and remote sensing reflectance calculation were carried out on the hyperspectral images of the oil spills obtained by the airborne Cubert-S185 sensor. The calculation is as follows.

$$R_{\mathrm{rs}} = \frac{L_{\mathrm{w}}}{\pi \cdot L_{\mathrm{p}}} \tag{2}$$

Here, $R_{\mathrm{rs}}$ represents the oil spill's remote sensing reflectance, $L_{\mathrm{w}}$ is the oil spill's radiometric data obtained by the S185 hyperspectral sensor, $L_{\mathrm{p}}$ represents the radiometric data of the AZ-WS20 calibration plate.

We used the 12-megapixel CMOS sensor mounted on the DJI-Mavic2 UAV to obtain high-resolution oil-spill optical images. We then performed a sample-assisted interpretation on the hyperspectral oil-spill images and produced the sample label data. We set the ratio of training samples to test samples to 1:9. The hyperspectral oil-spill images and labeled data for T1–T3 are shown in Figure 8.

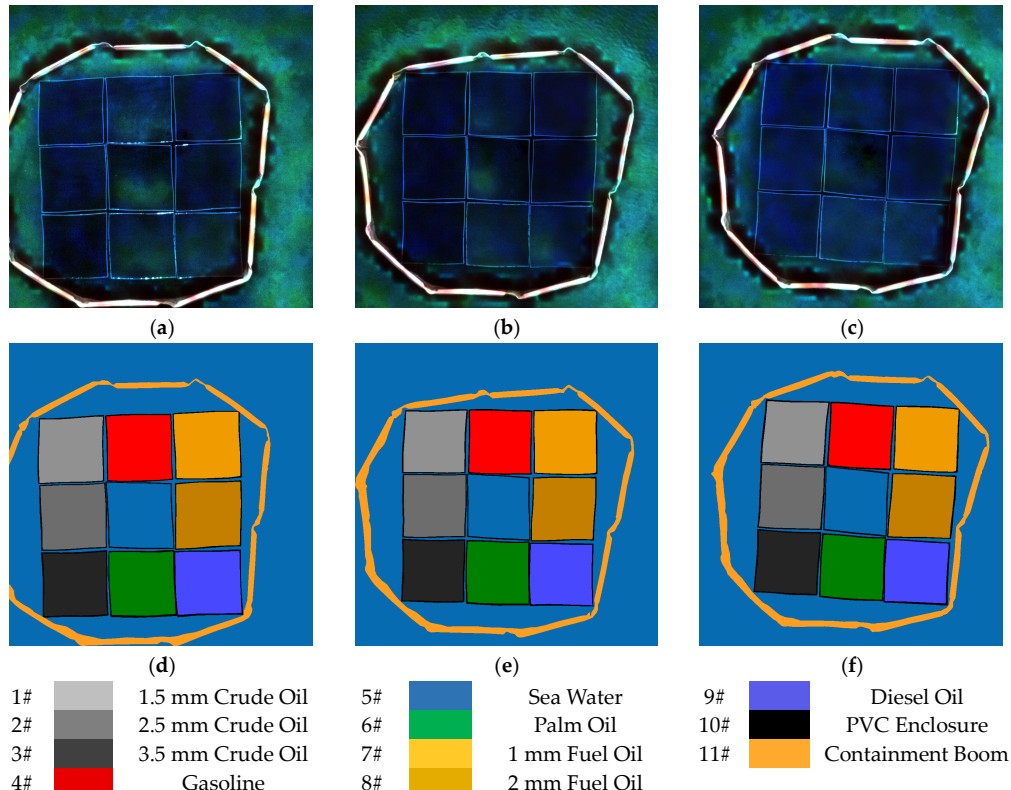

| 1# | | 1.5 mm Crude Oil | 5# | | Sea Water | 9# | | Diesel Oil |
| 2# | | 2.5 mm Crude Oil | 6# | | Palm Oil | 10# | | PVC Enclosure |
| 3# | | 3.5 mm Crude Oil | 7# | | 1 mm Fuel Oil | 11# | | Containment Boom |
| 4# | | Gasoline | 8# | | 2 mm Fuel Oil | | | |

**Figure 8.** Oil spill hyperspectral images (red: 482 nm, green: 550 nm, blue: 662 nm) and sample label data of T1-T3. (**a**,**d**) are image and label data at T1, respectively. (**b**,**e**) are image and label data at T2, respectively. (**c**,**f**) are image and label data at T3, respectively.

### 2.2. Adaptive Long-Term Moment Estimation Optimizer

The oil-spill data of the experiment were obtained using the UAV's hyperspectral sensor. Moreover, the data are $1000 \times 1000 \times 125$ pixels in size and have the characteristics of a high number of spectral dimensions and rich information. To ensure computational efficiency and training performance, the training of the deep learning models was carried out in multiple batches and over different periods.

ALTME improves the gradient update mechanism of the second momentum item and assigns an adaptive cumulative learning weight to the oil-spill spectral characteristic information of multiple batches and different periods. The adaptive selection of the second momentum item weight is realized through an iterative process, and the proportion of long-term accumulated spectral information learning is increased under the premise of ensuring model convergence. Deep learning experiments using the ALTME optimizer can theoretically enable the model to remember the long-term spectral characteristic information of different batches, allowing it to perform better and more smoothly during the iterative training; thus, improving the accuracy and stability of its oil-spill detection ability. In addition, we use a momentum correction term for the ALTME optimizer, which can effectively avoid instability in the first and second momentum terms due to the small gradient in the early stage of training, which will affect the model training.

The algorithm of the ALTME optimizer is shown in Algorithm 1, where $\beta$ represents the first moment, $\lambda$ is the adaptive cumulative learning weight, *iter* is the number of iterations, $g_t$ represents the gradient at time $t$, $m_t$ is the first momentum term, $v_t$ represents the second momentum term, $m_t^{correct}$ represents the first momentum correction term, $v_t^{correct}$ is the second momentum correction term, $Loss(\cdot)$ represents the loss function, $\theta_t$ is the weight of the deep learning model, $\alpha$ represents the learning rate, and $\varepsilon$ is the weight correction value.

---

**Algorithm 1** Adaptive Long-term Moment Estimation

---

1: **Input:** $\beta = 0.9$, $\lambda \in [\, 1 \times 10^{-4}, 1 \times 10^{-5}\,]$
2: **Initialize** $iter = 0$, $m_0 = 0$, $v_0 = 0$
3: **for** $\lambda = 1 \times 10^{-4}$ **to** $1 \times 10^{-5}$ **do**
4:　**Repeat**
5:　　**for** $t = 1$ **to** T **do**
6:　　　　$g_t = \nabla_\theta \, Loss(\,\theta_t\,)$
7:　　　　$m_t = \beta m_{t-1} + (\,1 - \beta\,)g_t$
8:　　　　$v_t = (\,1 + \lambda\,)v_{t-1} + \lambda g_t^2$
9:　　　　$m_t^{correct} = m_t \,/\, (\,1 - \beta^t\,)$
10:　　　　$v_t^{correct} = v_t \,/\, ((\,1 + \lambda\,)^t - 1\,)$
11:　　　　$\theta_{t} = \theta_{t-1} - \alpha \cdot m_t^{correct} \,/\, (\,\sqrt{v_t^{correct}} + \varepsilon\,)$
12:　　**End for**
13:　$iter \leftarrow iter + 1$
14: **End for**

---

We derive the gradient update process of the second momentum correction term $v_t^{correct}$ of the ALTME optimizers as follows.

$$
\begin{aligned}
v_t^{correct} &= \frac{v_t}{(\lambda+1)^t - 1} \\
&= \frac{(\lambda+1)v_{t-1} + \lambda g_t^2}{(\lambda+1)^t - 1} \\
&= \frac{(\lambda+1) \cdot \left[(\lambda+1)^{t-1} - 1\right]v_{t-1}^{correct} + \lambda g_t^2}{(\lambda+1)^t - 1} \\
&= \frac{(\lambda+1)^t - \lambda - 1}{(\lambda+1)^t - 1}v_{t-1}^{correct} + \frac{\lambda}{(\lambda+1)^t - 1}g_t^2 \\
&= \left[1 - \frac{\lambda}{(\lambda+1)^t - 1}\right]v_{t-1}^{correct} + \frac{\lambda}{(\lambda+1)^t - 1}g_t^2
\end{aligned}
\tag{3}
$$

We replace $1 - \frac{\lambda}{(\lambda+1)^t - 1}$ with $\varphi$, and the gradient updating process of the second momentum correction term becomes

$$
v_t^{correct} = \varphi v_{t-1}^{correct} + (1 - \varphi)g_t^2
\tag{4}
$$

The ALTME optimizer still updates the gradient based on the mechanism of exponential moving average. When $t \to \infty$, $\varphi \to 1$, and the learning rate will approach a constant. The second momentum term will update the model weights more smoothly with a smaller correction force so that the ALTME optimizer can not only learn the long-term spectral characteristic gradients of different training batches but also maintain better stability during the training and convergence stages.

### 2.3. Marine Oil-Spill Detection 1D-CNN Model

The aim of the proposed ALTME optimizer is to meet the needs of marine oil-spill detection based on hyperspectral remote sensing by conforming to the characteristics of experimental data and adapting to different oil-spill scenarios. Hence, by assigning an adaptive weight to the long-term spectral characteristic information of oil spills, we increase the cumulative learning proportion of the spectra of different batches to avoid the loss of effective spectral characteristic information.

Because the target of this detection research is an oil spill in an experimental enclosure, it has no obvious spatial characteristics. Therefore, the ALTME optimizer is implemented in a 1D-CNN, which is more suitable for extracting features from 1D spectral information to build the oil-spill detection model. Using this model, we extract the spectral characteristic data of the oil-spill hyperspectral images, construct the mapping relationship between

spectral characteristic data and the sample labels, and then identify the type of oil spills and detect oil films with different thicknesses. The model structure is shown in Figure 9.

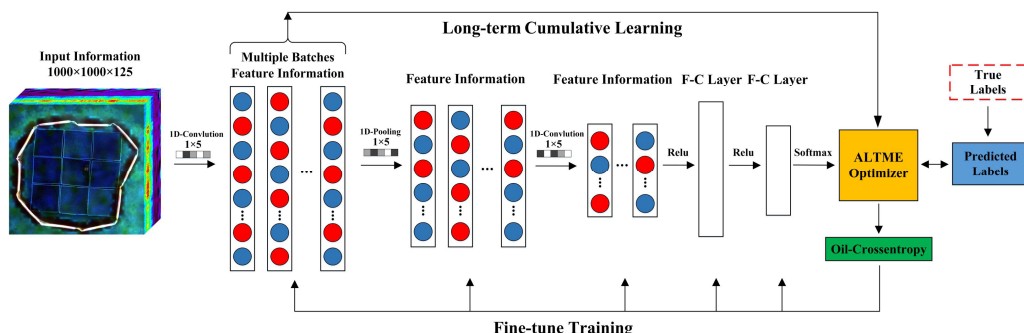

**Figure 9.** Structure of the marine oil spill detection 1D-CNN model.

In contrast to traditional convolution, the convolution in a 1D-CNN has a receptive field based on a 1D plane convolution kernel, which has a strong feature extraction ability and can fit nonlinear data well. Because of the sparse representation and weight sharing, the number of parameters of the marine oil-spill detection model is greatly reduced, and the efficiency of detection is improved. The calculation is as follows.

$$h_{i,j} = g\left[\sum_{m}^{M}\left(h_{i-1,m} \times w_{i,mj}\right) + b_{i,j}\right] \tag{5}$$

Here, $h_{i,j}$ represents the $j$th output feature map of the $i$th convolution layer, $M$ is the number of feature maps of the $i$th convolution layer, $w_{i,mj}$ represents the weight, $b_{i,j}$ is the bias, and $g$(f) represents the activation function.

In the model proposed in this paper, the rectified linear unit (ReLU) function is chosen as the activation function. The function enables the model to perform gradient descent more efficiently while maintaining a fast calculation speed and can avoid the phenomena of gradient disappearance and expansion. The formula of the ReLU function is

$$g(x) = \max(0, x) \tag{6}$$

We set the pooling method of the detection model to 1D maximum pooling, which can reduce the risk of overfitting and enhance the robustness of the model through the pooling process. Each pooling layer corresponds to the receptive field of the convolutional layer of size $N \times 1$. The maximum pooling formula is as follows.

$$a_j = \max_{N \times 1}\left(a_i^{n \times 1} u(n, 1)\right) \tag{7}$$

Here, max() represents the 1D maximum pooling function, $u(n, 1)$ is the window function of the convolution layer, and $a_j$ is the maximum value in the neighborhood.

Since the research is oriented to the detection of multiple oil spill targets, multi-classification cross entropy is selected as the loss function to carry out oil spill detection based on a softmax classifier. The oil-crossentropy function is as follows.

$$\mathrm{oil - crossentropy} = -\left[\sum_{i=1}^{n} P_i \cdot \log\left(\frac{e^{Z_i}}{Z}\right)\right], Z = \sum_{i=1}^{n} e^{z_i} \tag{8}$$

Here, $Z_i$ is the logits function of the softmax classifier, $n$ is the number of detection targets, $P_i$ is the true sample distribution.

The reverse fine-tuning process of the oil-spill detection model uses the backpropagation algorithm combined with the calibrated label data to adjust the weight and parameters

of the model layer by layer so that the model can optimize the mapping of the oil-spill spectral characteristic data.

The structure of the proposed model, which is based on a Keras and TensorFlow framework, is shown in Table 4. The model consists of three 1D convolutional layers, three 1D maximum pooling layers, and two fully connected layers. Through iterative training and hyperparameter tuning, the model acquires the ability to identify oil-spill types and detect oil films with different thicknesses.

**Table 4.** Structure of the marine oil spill detection 1D-CNN model.

| Layer | Number | Kernel | Stride |
|---|---|---|---|
| Convolutional layer-1 | 150 | $1 \times 5$ | 1 |
| MaxPooling layer-1 | - | $1 \times 5$ | 3 |
| Convolutional layer-2 | 100 | $1 \times 5$ | 1 |
| MaxPooling layer-2 | - | $1 \times 5$ | 3 |
| Convolutional layer-3 | 50 | $1 \times 5$ | 1 |
| MaxPooling layer-3 | - | $1 \times 5$ | 1 |
| Fully connected layer-1 | 200 | - | - |
| Fully connected layer-2 | 100 | - | - |

*2.4. Accuracy Evaluation Indexes*

The overall accuracy (OA) and Kappa coefficient were respectively chosen to measure the overall detection performance and consistency of the oil-spill detection results. Because the seawater pixels in the oil-spill hyperspectral images account for a large proportion of the image and have a greater impact on OA, the OA alone cannot effectively reflect the detection accuracy of each experimental group. By contrast, the $F_1$-score takes into account both recall and precision, which removes the imbalance between the two metrics. Therefore, the $F_1$-score was selected to measure the single-target detection accuracy of the proposed model for each experimental group. The calculation formulas of each accuracy evaluation index are as follows:

$$\text{Recall} = \frac{\text{TP}}{\text{TP} + \text{FN}} \times 100\% \tag{9}$$

$$\text{Precision} = \frac{\text{TP}}{\text{TP} + \text{FP}} \times 100\% \tag{10}$$

$$F_1 - \text{score} = 2 \cdot \frac{\text{Recall} \cdot \text{Precision}}{\text{Recall} + \text{Precision}} \tag{11}$$

$$\text{OA} = \frac{\text{TP} + \text{TN}}{\text{TN} + \text{TP} + \text{FN} + \text{FP}} \times 100\% \tag{12}$$

$$\text{Kappa} = \frac{\text{TP} + \text{TN}}{(\text{TN} + \text{FP}) \cdot (\text{TN} + \text{FN}) \cdot (\text{FN} + \text{TP}) \cdot (\text{FP} + \text{TP})} \tag{13}$$

where TP represents the number of true positives, TN is the number of true negatives, FP represents the number of false positives, and FN is the number of false negatives.

**3. Results and Discussion**

*3.1. Analysis of Oil-Spill Spectrum Characteristics*

We averaged and smoothed the hyperspectral data of the oil spill and seawater experimental groups. The remote sensing reflectance hyperspectral curves of the experimental groups at T1–T3 are shown in Figures 10–12, respectively.

3.1.1. Seawater and Light-Oil Experimental Groups at T1

The remote sensing reflectance of the seawater (group 5) is higher than all the oil-spill groups in the visible light range of 450–720 nm. In the near-infrared range beyond 760 nm, the reflectance decreases because of the absorption characteristics of seawater. Because of the chlorophyll in the water, the hyperspectral curve of seawater has a peak at the

560 nm green light band. The reflectance trends of the three light-oil experimental groups of gasoline (group 4), palm oil (group 6), and diesel (group 9) are similar, and the overall reflectance is lower than that of seawater. Because of the high transmittance of light oils, the three groups of light oils are strongly affected by the background water body, and the spectral curves all have small peaks around 560 nm. In addition, the reflectances of the three light-oil experimental groups are not very separable in the whole band, and there is little information that could be extracted and poor potential for type recognition.

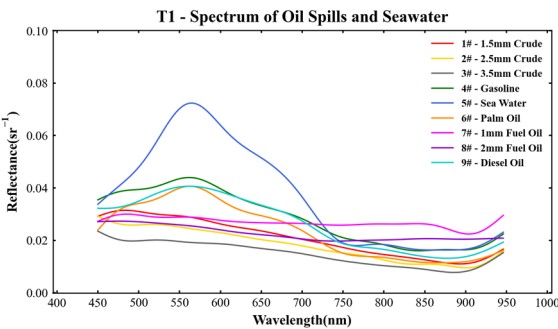

**Figure 10.** Remote sensing reflectance of the oil spills at T1.

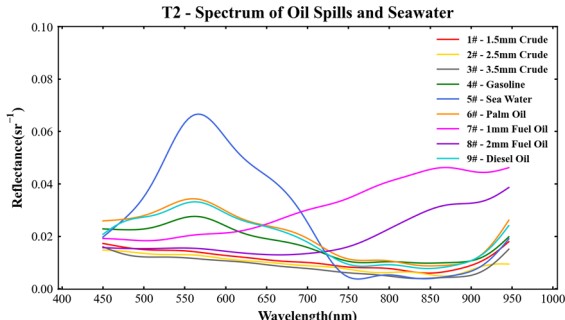

**Figure 11.** Remote sensing reflectance of the oil spills at T2.

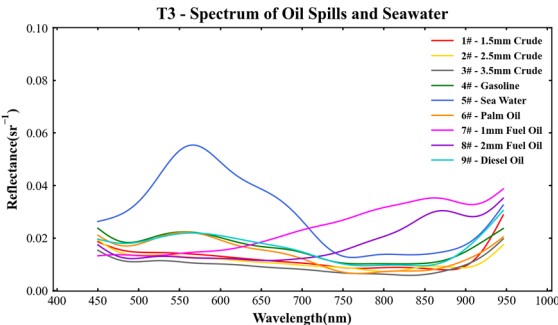

**Figure 12.** Remote sensing reflectance of the oil spills at T3.

3.1.2. Crude Oil Experimental Groups at T1

The spectral curves of the crude oil (groups 1–3) with different thicknesses are spectrally separable in the visible and near-infrared bands of 450–900 nm. The overall spectrum shows a downward trend, and the overall reflectance is inversely proportional to the oil film thickness, which is consistent with Liu and Sun's research conclusion [46,47]. We believe that this is related to the strong absorption characteristics of crude oil. The thicker the oil film, the stronger the absorption capacity of sunlight, which decreases the reflectance. The reflectance of these groups tends to increase in the near-infrared bands of 900-950 nm, which is related to the backscattering properties of crude oil.

### 3.1.3. Fuel Oil Experimental Groups at T1

The remote sensing reflectance curves of the fuel oil (groups 7 and 8) with different thicknesses tend to be relatively gentle over the whole waveband, and there is spectral separability information that can be extracted. The overall reflectance is inversely proportional to the oil film thickness. Moreover, the overall reflectance of the 1-mm thick fuel oil (group 7) is higher than that of the 2-mm thick fuel oil (group 8). The oil film spectrum curve of group 7 dips at 910 nm. Because of the backscattering effect of fuel oil, the curves of these groups tend to increase after 910 nm.

### 3.1.4. Seawater and Light-Oil Experimental Groups at T2–T3

As time passes, the solar elevation angle decreases, the sunlight intensity weakens, and the overall remote sensing reflectance of the seawater (group 5) gradually decreases. The reflectance of the seawater is still affected by the chlorophyll at T2–T3, so the reflection peak appears at 560 nm, and the reflectance decreases in the near-infrared bands. The reflectance of the three light-oil experimental groups is affected by the background water body, and there is still a reflection peak at 560 nm, but with the passage of time, this peak gradually decreases. In addition, as sunlight intensity decreases, the spectral separability of the light-oil groups further decreases. In particular, at T3, the reflectance of the light-oil groups has only a weak spectral separability in the visible light bands.

### 3.1.5. Crude Oil Experimental Groups at T2–T3

Because the sunlight intensity has decreased, the separability of the spectral curves of the crude oil with different thicknesses (groups 1–3) in the range of 450–900 nm is lower than that at T1. However, the overall remote sensing reflectance still retains the overall inverse proportionality with respect to the oil film thickness; that is, a thicker oil film thickness leads to a lower overall reflectance. The reflectances of the crude oil in groups 1–3 tend to increase in the near-infrared bands of 900–950 nm, which is more obvious than at T1. We believe that over time, crude oil and seawater may undergo partial emulsification, causing the reflectance of these groups to increase in the near-infrared bands.

### 3.1.6. Fuel Oil Experimental Groups at T2–T3

The reflectance curves of the fuel oil (groups 7 and 8) change substantially when compared with the results from T1. Moreover, the spectral response begins to appear in the visible light bands, and the overall waveband tends to increase. We believe this phenomenon is caused by the emulsification reactions of fuel oil and seawater. In addition, groups 7 and 8 still have spectral separability, the overall reflectance is inversely proportional to the oil film thickness, and both experimental groups have spectral troughs at the 910-nm band.

In summary, the spectral curves of the light-oil experimental groups are similar, and the spectral separability is weak. As time passes from T1 to T3, the sunlight intensity weakens and the spectral separability further decreases, and only a weak spectral difference exists at T3. Moreover, there are certain spectral differences between the groups of heavy oil with different thicknesses, and the spectral separability decreases as sunlight intensity decreases. When sunlight intensity is low, it may be difficult to accurately distinguish the type of light oil and detect the thickness of heavy oils based on traditional remote sensing methods. Therefore, we apply deep learning techniques to the actual measured hyperspectral oil-spill data to effectively extract the characteristic information of oil-spill spectral curves to accurately identify the type of oil spills and detect the thickness of the thick oil films.

### 3.2. Selection and Analysis of Basic Oil-Spill Detection Models

Because this detection experiment is for 1D oil-spill spectral information, we built representative SVM, GRU, and 1D-CNN oil-spill detection models using Keras and TensorFlow, which are widely used in 1D signal recognition. We used the oil-spill UAV hyperspectral

data and compared and analyzed the detection results obtained by different models at different times and phases, and selected the best basic marine oil-spill detection model. All experiments were performed on a Dell Precision 7820 Tower Workstation equipped with a Xeon-Gold 5122 processor and an NVIDIA-RTX2080Ti graphics processing unit. Each group of oil-spill detection experiments was carried out five times, and the reported detection accuracy is the average value of the five experiments, as shown in Tables 5–7.

**Table 5.** Accuracy of the detection results at T1.

| | | Index | | | | | | | |
|---|---|---|---|---|---|---|---|---|---|
| **Model** | | **SVM** | | | **GRU** | | | **1D-CNN** | |
| **Group** | **Recall** | **Precision** | $F_1$-**Score** | **Recall** | **Precision** | $F_1$-**Score** | **Recall** | **Precision** | $F_1$-**Score** |
| 1#-1.5 mm Crude | 85.93 | 83.88 | 0.849 | 90.07 | 95.98 | 0.929 | 98.75 | 96.42 | **0.976** |
| 2#-2.5 mm Crude | 78.95 | 79.33 | 0.791 | 93.98 | 89.84 | 0.919 | 97.40 | 97.73 | **0.976** |
| 3#-3.5 mm Crude | 85.84 | 83.08 | 0.844 | 93.41 | 95.52 | 0.945 | 97.05 | 98.07 | **0.976** |
| 4#-Gasoline | 87.45 | 86.89 | 0.872 | 92.80 | 89.84 | 0.913 | 98.55 | 97.62 | **0.981** |
| 5#-Seawater | 96.82 | 95.57 | 0.962 | 97.75 | 97.82 | 0.978 | 98.48 | 98.63 | **0.986** |
| 6#-Palm Oil | 84.65 | 87.21 | 0.859 | 95.43 | 91.79 | 0.936 | 99.03 | 95.31 | **0.971** |
| 7#-1 mm Fuel Oil | 93.48 | 89.47 | 0.914 | 98.32 | 97.48 | 0.979 | 98.27 | 98.56 | **0.984** |
| 8#-2 mm Fuel Oil | 90.26 | 95.47 | 0.928 | 95.56 | 97.79 | 0.967 | 98.52 | 97.02 | **0.978** |
| 9#-Diesel Oil | 82.14 | 92.21 | 0.869 | 94.61 | 92.34 | 0.935 | 98.16 | 97.45 | **0.978** |
| 10#-PVC | 79.40 | 86.76 | 0.829 | 82.55 | 88.88 | 0.856 | 85.52 | 89.13 | **0.873** |
| 11#-Oil Boom | 92.8 | 94.93 | 0.939 | 95.58 | 94.51 | **0.951** | 93.51 | 96.39 | 0.949 |
| OA (%) | | 92.34 | | | 95.97 | | | **97.76** | |
| Kappa | | 0.882 | | | 0.939 | | | **0.966** | |
| Time (min) | | 19.12 | | | **4.79** | | | 11.68 | |

**Table 6.** Accuracy of the detection results at T2.

| | | Index | | | | | | | |
|---|---|---|---|---|---|---|---|---|---|
| **Model** | | **SVM** | | | **GRU** | | | **1D-CNN** | |
| **Group** | **Recall** | **Precision** | $F_1$-**Score** | **Recall** | **Precision** | $F_1$-**Score** | **Recall** | **Precision** | $F_1$-**Score** |
| 1#-1.5 mm Crude | 71.92 | 80.13 | 0.758 | 85.12 | 89.73 | 0.874 | 95.66 | 98.05 | **0.968** |
| 2#-2.5 mm Crude | 85.77 | 76.07 | 0.806 | 97.23 | 88.65 | 0.927 | 98.55 | 95.76 | **0.971** |
| 3#-3.5 mm Crude | 66.77 | 82.21 | 0.737 | 83.45 | 89.70 | 0.865 | 96.00 | 97.23 | **0.966** |
| 4#-Gasoline | 80.36 | 78.95 | 0.797 | 92.40 | 88.65 | 0.905 | 97.56 | 96.30 | **0.969** |
| 5#-Seawater | 97.68 | 94.80 | 0.962 | 97.56 | 97.61 | 0.976 | 98.30 | 98.62 | **0.985** |
| 6#-Palm Oil | 83.90 | 81.74 | 0.828 | 95.96 | 87.08 | 0.913 | 98.72 | 93.24 | **0.959** |
| 7#-1 mm Fuel Oil | 97.06 | 97.10 | 0.971 | 98.41 | 97.60 | 0.980 | 98.28 | 98.84 | **0.986** |
| 8#-2 mm Fuel Oil | 96.66 | 96.69 | 0.967 | 97.79 | 97.37 | 0.976 | 98.92 | 98.32 | **0.986** |
| 9#-Diesel Oil | 71.93 | 87.68 | 0.790 | 88.48 | 93.74 | 0.910 | 97.80 | 94.46 | **0.961** |
| 10#-PVC | 76.00 | 92.73 | 0.835 | 80.99 | 90.31 | 0.854 | 83.08 | 91.09 | **0.869** |
| 11#-Oil Boom | 94.84 | 95.93 | 0.954 | 96.14 | 95.92 | **0.960** | 96.27 | 95.56 | 0.959 |
| OA (%) | | 91.76 | | | 95.31 | | | **97.53** | |
| Kappa | | 0.869 | | | 0.927 | | | **0.961** | |
| Time (min) | | 18.15 | | | **4.95** | | | 11.77 | |

As Table 5 shows, in the oil-spill detection experiment at time T1, the OA of the 1D-CNN model reaches 97.76%, and the Kappa coefficient reaches 0.966, both of which are higher than the detection accuracies of the SVM and GRU models. The 1D-CNN model has higher detection accuracy $F_1$-scores for the seawater (group 5), oil spills (groups 1–4 and 6–9), and PVC enclosure (group 10) than the SVM and GRU models. The recognition accuracy $F_1$-score for light-oil types is more than 0.971, and the detection accuracy for heavy oils with different thicknesses is more than 0.976. The GRU model is more sensitive to the detection of the oil containment boom (group 11), and its $F_1$-score reaches 0.951, which is better than the scores of the other two detection models. In addition, the calculation speed

of the GRU model is much higher than that of the SVM and 1D-CNN models; its oil-spill detection process only takes 4.79 min.

**Table 7.** Accuracy of the detection results at T3.

| | | Index | | | | | | | |
|---|---|---|---|---|---|---|---|---|---|
| Model | | SVM | | | GRU | | | 1D-CNN | |
| Group | Recall | Precision | $F_1$-Score | Recall | Precision | $F_1$-Score | Recall | Precision | $F_1$-Score |
| 1#-1.5 mm Crude | 74.79 | 79.05 | 0.769 | 88.38 | 94.64 | 0.914 | 98.40 | 96.44 | **0.974** |
| 2#-2.5 mm Crude | 79.48 | 65.54 | 0.718 | 92.05 | 87.19 | 0.896 | 98.37 | 96.09 | **0.972** |
| 3#-3.5 mm Crude | 53.16 | 76.10 | 0.626 | 81.36 | 87.59 | 0.844 | 95.10 | 97.36 | **0.962** |
| 4#-Gasoline | 71.69 | 70.41 | 0.710 | 91.74 | 88.54 | 0.901 | 96.76 | 95.94 | **0.964** |
| 5#-Seawater | 97.79 | 93.95 | 0.958 | 97.75 | 97.15 | 0.975 | 98.52 | 98.36 | **0.984** |
| 6#-Palm Oil | 74.03 | 78.47 | 0.762 | 95.57 | 88.74 | 0.920 | 98.26 | 94.86 | **0.965** |
| 7#-1 mm Fuel Oil | 95.60 | 96.17 | 0.959 | 97.59 | 96.24 | 0.969 | 98.34 | 97.75 | **0.981** |
| 8#-2 mm Fuel Oil | 91.79 | 95.20 | 0.935 | 95.36 | 96.67 | 0.960 | 98.52 | 96.96 | **0.977** |
| 9#-Diesel Oil | 58.73 | 66.98 | 0.626 | 89.78 | 86.85 | 0.883 | 96.90 | 94.00 | **0.954** |
| 10#-PVC | 67.24 | 95.08 | 0.788 | 71.85 | 93.22 | 0.812 | 73.94 | 93.22 | **0.825** |
| 11#-Oil Boom | 95.57 | 96.55 | 0.961 | 97.07 | 95.48 | **0.963** | 96.89 | 95.38 | 0.961 |
| OA (%) | | 89.23 | | | 94.84 | | | **97.32** | |
| Kappa | | 0.829 | | | 0.920 | | | **0.958** | |
| Time (min) | | 17.08 | | | **4.88** | | | 11.73 | |

The results in Table 6 show that in the oil-spill detection experiment at T2, the 1D-CNN model performs well, with an OA of 97.53% and a Kappa coefficient of 0.961, both of which are higher than the SVM and GRU models. The $F_1$-scores of the 1D-CNN model for the seawater (group 5), the oil spill (groups 1–4 and 6–9), and the enclosure (group 10) are overall higher than those of the SVM and GRU models. The recognition accuracy $F_1$-score for light-oil types is larger than 0.959, and the detection accuracy for heavy-oil groups with different thicknesses is larger than 0.966. The detection accuracy $F_1$-score of the GRU model for the oil containment boom (group 11) is 0.960, which is higher than the scores of the other two models, indicating that this model retains excellent detection performance.

It can be seen from Table 7 that in the detection experiment at T3, the OA of the 1D-CNN reaches 97.32%, and the Kappa coefficient reaches 0.958, both of which are higher than the scores of the other two types of detection models. The 1D-CNN model has a higher detection accuracy $F_1$-score for all single targets except the oil containment boom (group 11), but its detection speed is slower, taking 11.73 min. The detection accuracy $F_1$-score for heavy oils with different thicknesses is larger than 0.962, and the recognition accuracy for light-oil types is larger than 0.954. The GRU model still detects the oil containment boom (group 11) best, with an $F_1$-score of 0.963. In addition, the GRU model still detects oil spills very efficiently, taking only 4.88 min, which is much faster than the other two models.

As shown in Figure 13, in the detection results of the SVM model at time T1, there are obvious misclassifications in several experimental groups. Moreover, the SVM detection model has a generally poor detection performance for the experimental groups of heavy oils with different thicknesses, and there is much room for improvement in the identification of light oils. The detection result of the GRU model at T1 is better than that of SVM, and the detection performance of the fuel oil (groups 7 and 8) is improved, but there is a partial misclassification in the 3.5-mm thick crude oil (group 3) and the seawater (group 5). Compared with the other two types of oil-spill detection models, the 1D-CNN model has a better overall detection performance, but the problem of misclassification in the 3.5-mm thick crude oil (group 3) and the fuel oil (groups 7 and 8) has not been completely resolved.

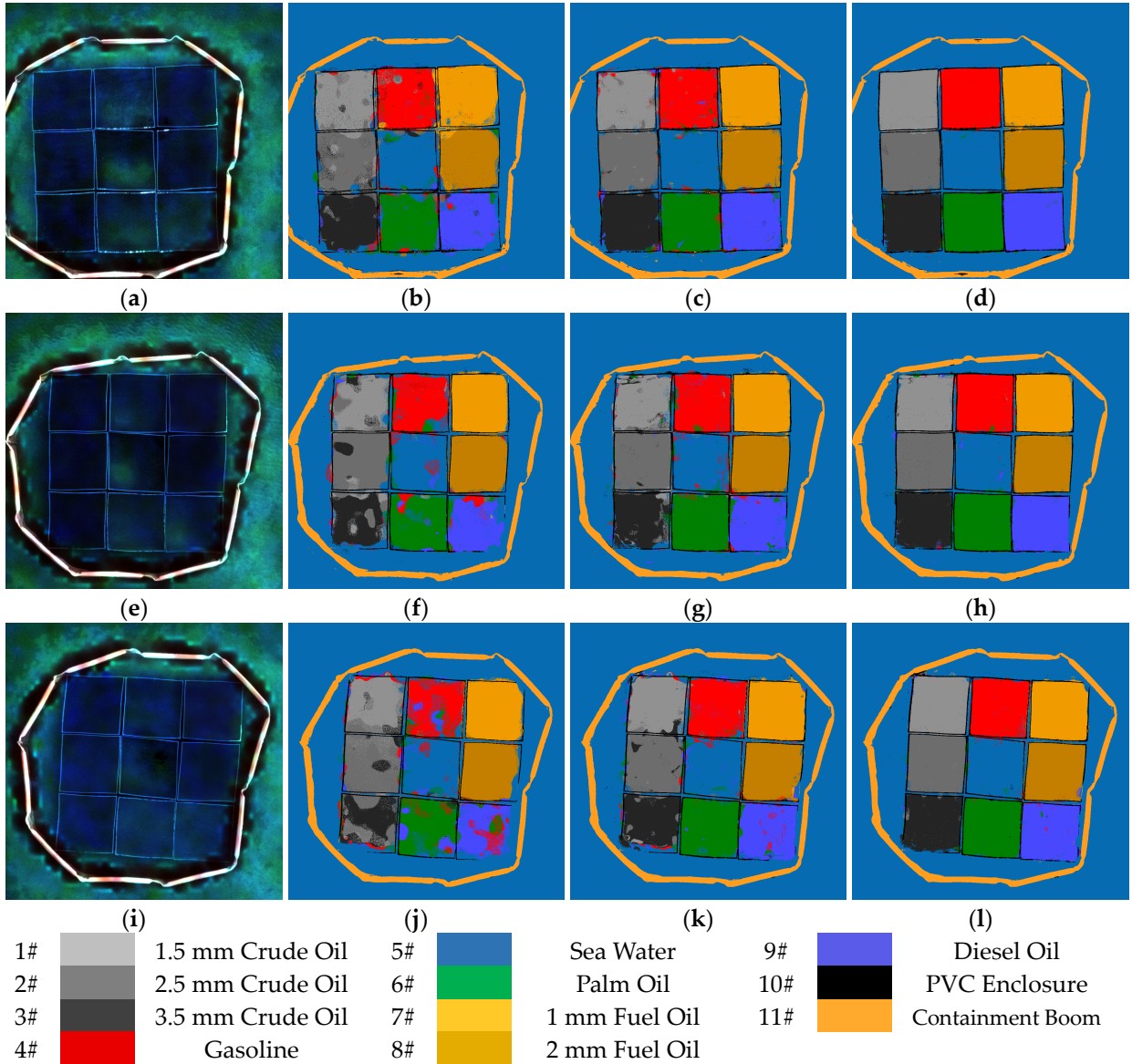

| 1# | | 1.5 mm Crude Oil | 5# | | Sea Water | 9# | | Diesel Oil |
|---|---|---|---|---|---|---|---|---|
| 2# | | 2.5 mm Crude Oil | 6# | | Palm Oil | 10# | | PVC Enclosure |
| 3# | | 3.5 mm Crude Oil | 7# | | 1 mm Fuel Oil | 11# | | Containment Boom |
| 4# | | Gasoline | 8# | | 2 mm Fuel Oil | | | |

**Figure 13.** Detection results of the marine oil spill detection model. (**a**,**e**,**i**) are true color images at T1–T3, respectively (red: 482 nm, green: 550 nm, blue: 662 nm). (**b**,**f**,**j**) are detection results of the SVM model at T1–T3, respectively. (**c**,**g**,**k**) are detection results of the GRU model at T1–T3, respectively. (**d**,**h**,**l**) are detection results of the 1D-CNN model at T1–T3, respectively.

At T2, the detection results of the SVM and GRU models for the fuel oil (groups 7 and 8) are acceptable. However, their detection performance for crude oil (groups 1–3), gasoline (group 4), seawater (group 5), and diesel oil (group 9) are poor. These models fail to complete the tasks of detecting the thickness of the heavy-oil film and identifying the type of oil spill, and neither of them could effectively detect the oil containment boom of the diesel experiment (group 9). Compared with the other two models, the detection performance of the 1D-CNN model at time T2 is much better, but there is still room for improvement in the identification of the enclosures for the gasoline (group 5) and diesel (group 9).

At T3, the SVM and GRU oil-spill detection models have more difficulty than they did at T1 and T2, and they cannot effectively extract the differences between the oil-spill spectra and hence cannot identify the type of oil spills and detect the thickness of the oil films. The detection result of the 1D-CNN model at T3 is better than those of the other two models, but the overall detection performance is worse than it is at T1 and T2. The recognition

results of the model on the enclosures of the 3.5-mm thick crude oil (group 3), gasoline (group 5), 2-mm thick fuel oil (group 8), and diesel (group 9) still need to be improved.

In summary, the OA and Kappa coefficients of the 1D-CNN oil-spill detection model at T1–T3 are higher than those of the SVM and GRU models, and its single target detection accuracy $F_1$-scores for different oil-spill experimental groups are all higher than the other two detection models. From the perspective of detection performance, at each observation point, the 1D-CNN model is better than the SVM and GRU oil-spill detection models. The advantage of the GRU model is that the model converges quickly and can accurately identify the oil containment boom. Moreover, the oil-spill detection process only takes about 4 min. In the future, it could be used for rapid emergency detection in large-scale marine oil-spill disasters. Under the same experimental conditions, the detection time of the 1D-CNN model is about 12 min, but the detection accuracy and performance of this model are much better than those of the GRU model, and the calculation efficiency is within an acceptable range for this experiment.

In addition, we found that the detection performance of the same oil-spill detection model at different moments deteriorates over time, and the detection performances of the models at T3 are substantially worse than they are at T1 and T2. We believe that this is related to the solar elevation angle, which decreases over time in this experiment, and the sunlight intensity, which correspondingly decreases. It is not difficult to see in Figures 10–12 that, from T1 to T3, as the sunlight intensity gradually decreases, the spectral separability between different types of light oils and heavy oils with different thicknesses weakens. Hence, the detection performance of an oil-spill detection model decreases over time. The experimental results show that the reduction in sunlight intensity reduces the spectral separability of the oil-spill experimental groups, which has a substantial impact on the SVM and GRU models, whereas the 1D-CNN model yields relatively stable detection capabilities. In this study, based on the detection accuracy, performance, and single-target accuracy of the three oil-spill detection models at different times, the 1D-CNN model was selected as the basic oil-spill detection model.

### 3.3. Adaptive Selection and Analysis of the Cumulative Learning Weight of the ALTME Optimizer

The ALTME optimizer assigns adaptive cumulative learning weights to the long-term spectral information of experimental hyperspectral data, making full use of the oil-spill spectral information and achieving the purpose of cumulatively learning the spectral characteristic information of different batches. We have determined through experiments that the adaptive range of the cumulative learning weight $\lambda$ of the model under the premise of stable convergence is $1 \times 10^{-4}$–$5 \times 10^{-4}$. We then used iterative experiments to realize the adaptive selection process of the cumulative learning weight. The optimizer considers the OA and Kappa coefficients, compares the $F_1$-score of different oil-spill groups, and extracts the spectral cumulative learning weights that best fit the oil-spill experiment scene at different times. The ALTME optimizer was then constructed using the selected cumulative learning weight, which was integrated with the 1D-CNN to build the marine oil-spill detection 1D-CNN model for identifying the type of oil spills and detecting heavy-oil films with different thicknesses. As shown in Table 8, at T1–T3, the adaptive cumulative learning weight $\lambda$ of the ALTME optimizer has values of $3 \times 10^{-4}$, $3 \times 10^{-4}$, and $2 \times 10^{-4}$, respectively.

As Table 8 shows, the OA and Kappa coefficients of the marine oil-spill detection model at T1 are higher than those at T2 and T3, and the accuracy of detection for all single targets except for fuel oil (groups 7 and 8) is also higher. The recognition accuracy $F_1$-score for light-oil types is larger than 0.972, and the detection accuracy for heavy oils is larger than 0.980. Experimental results show that although the oil-spill detection model has strong feature extraction capabilities, it is limited by the low spectral difference of the oil-spill spectral data at T2 and T3, which will affect detection accuracy to a certain extent. It can be seen from Figure 14 that after the oil-spill detection model has been equipped with the

ALTME optimizer, the detection capability is fully enhanced at T1–T3, and the OA, Kappa, and single detection accuracy $F_1$-score values have all been improved to varying degrees.

**Table 8.** Accuracy of the adaptive selection of the cumulative learning weight at T1-T3.

| | Index | | | | | | | | |
|---|---|---|---|---|---|---|---|---|---|
| **Moment** | **T1** | | | **T2** | | | **T3** | | |
| $\lambda$ | $3 \times 10^{-4}$ | | | $3 \times 10^{-4}$ | | | $2 \times 10^{-4}$ | | |
| **Group** | **Recall** | **Precision** | **$F_1$-Score** | **Recall** | **Precision** | **$F_1$-Score** | **Recall** | **Precision** | **$F_1$-Score** |
| 1#-1.5 mm Crude | 99.42 | 99.53 | **0.995** | 99.28 | 99.54 | 0.994 | 99.14 | 97.59 | 0.984 |
| 2#-2.5 mm Crude | 99.56 | 99.08 | **0.993** | 99.63 | 98.67 | 0.991 | 98.91 | 97.78 | 0.983 |
| 3#-3.5 mm Crude | 99.49 | 98.85 | **0.992** | 99.58 | 98.13 | 0.989 | 98.29 | 97.69 | 0.980 |
| 4#-Gasoline | 99.46 | 99.19 | **0.993** | 99.44 | 98.61 | 0.990 | 98.82 | 97.44 | 0.981 |
| 5#-Seawater | 99.42 | 99.31 | **0.994** | 99.32 | 99.27 | 0.993 | 98.89 | 98.90 | 0.989 |
| 6#-Palm Oil | 99.67 | 99.11 | **0.994** | 99.85 | 97.92 | 0.989 | 98.52 | 98.38 | 0.985 |
| 7#-1 mm Fuel Oil | 99.66 | 99.30 | 0.995 | 99.74 | 99.39 | **0.996** | 98.78 | 98.65 | 0.987 |
| 8#-2 mm Fuel Oil | 99.69 | 98.67 | 0.992 | 99.65 | 99.38 | **0.995** | 99.35 | 97.52 | 0.984 |
| 9#-Diesel Oil | 99.40 | 98.99 | **0.992** | 99.50 | 98.66 | 0.991 | 98.88 | 95.56 | 0.972 |
| 10#-PVC | 86.84 | 92.52 | 0.896 | 87.04 | 93.32 | **0.901** | 77.93 | 92.13 | 0.844 |
| 11#-Oil Boom | 96.80 | 97.62 | **0.972** | 96.37 | 97.90 | 0.971 | 97.11 | 95.75 | 0.964 |
| OA (%) | | **98.96** | | | 98.35 | | | 98.09 | |
| Kappa | | **0.984** | | | 0.974 | | | 0.970 | |
| Time (min) | | **12.15** | | | 12.22 | | | 12.07 | |

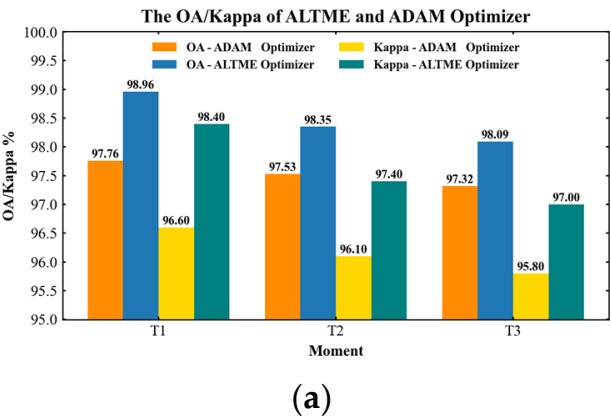

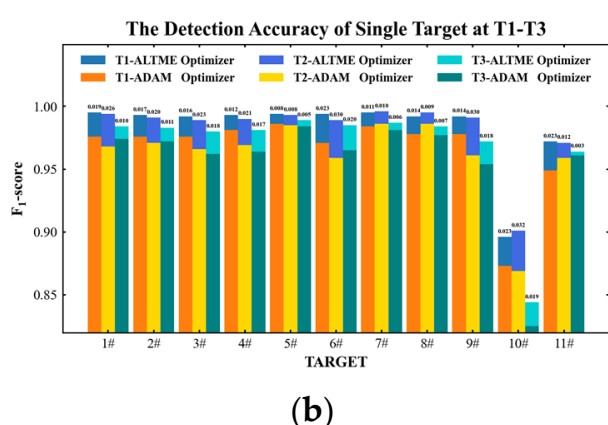

**(a)**                   **(b)**

**Figure 14.** Comparison of the detection accuracy of the marine oil spill detection model with the ALTME and Adam optimizers. (**a**) OA and Kappa coefficient of the detection results and (**b**) single accuracy $F_1$-score of the detection results.

It can be seen from Figure 15 that at T1, the marine oil-spill detection model equipped with the ALTME optimizer can accurately identify the type of oil spills and detect different thicknesses of thick oil films, correcting the obvious misclassifications of the basic oil-spill detection 1D-CNN model for 3.5-mm thick crude oil (group 3) and fuel oil (groups 7 and 8). At T2, the proposed detection model improves the detection ability of the basic model for the 3.5-mm thick crude oil (group 3) and improves the identification performance of gasoline (group 5) and diesel (group 9). At T3, the proposed detection model counters the influence of poor data spectral separability. By cumulatively learning long-term oil-spill spectral characteristic information, the problem of misclassifications of the experimental enclosure is effectively solved, and the overall oil-spill detection results are improved.

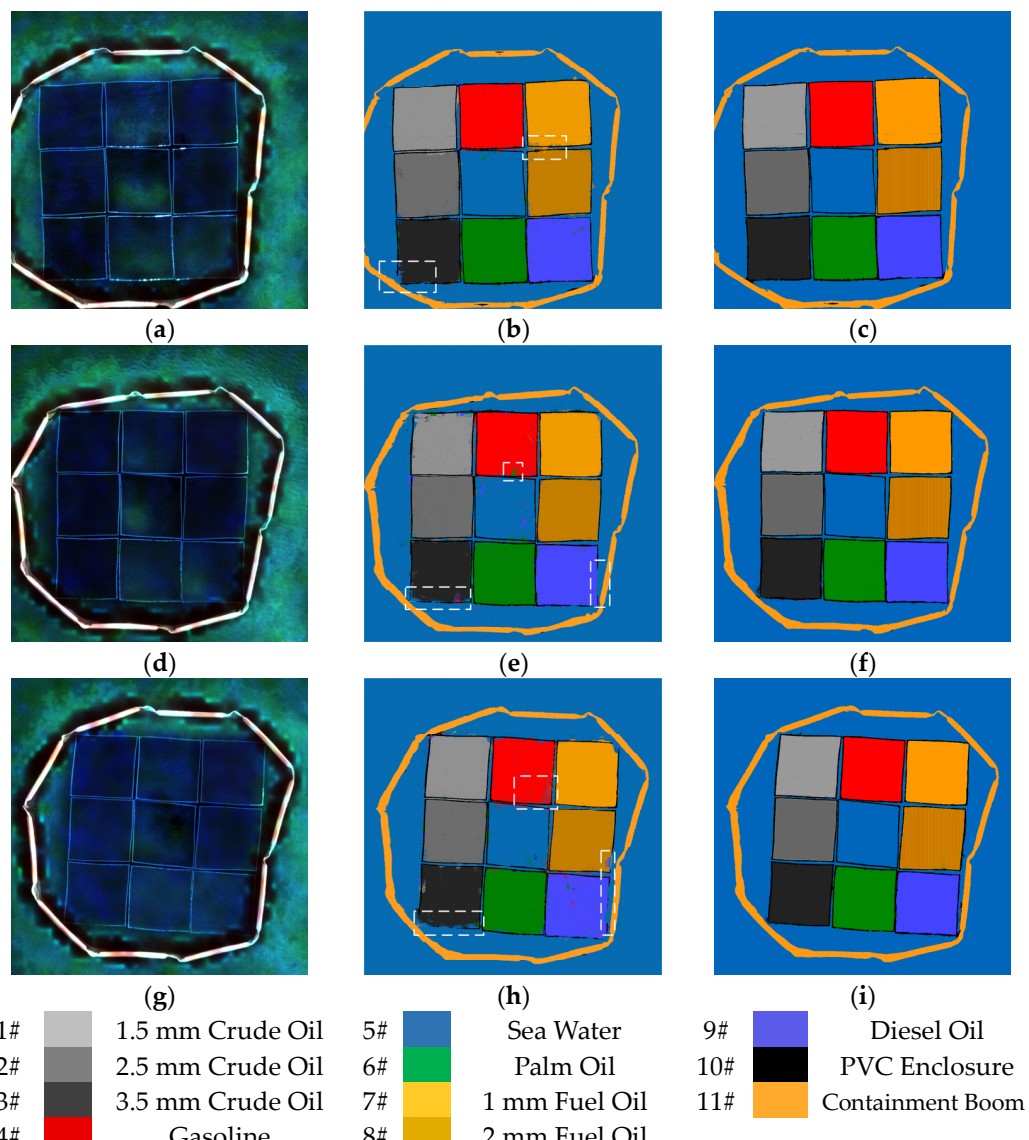

| | | | | | |
|---|---|---|---|---|---|
| 1# | 1.5 mm Crude Oil | 5# | Sea Water | 9# | Diesel Oil |
| 2# | 2.5 mm Crude Oil | 6# | Palm Oil | 10# | PVC Enclosure |
| 3# | 3.5 mm Crude Oil | 7# | 1 mm Fuel Oil | 11# | Containment Boom |
| 4# | Gasoline | 8# | 2 mm Fuel Oil | | |

**Figure 15.** Detection results of the oil spill detection model with the ALTME optimizer. (**a,d,g**) represent the true color images at T1–T3, respectively (red: 482 nm, green: 550 nm, blue: 662 nm). (**b,e,h**) are detection results of the basic 1D-CNN model at T1–T3, respectively. (**c,f,i**) are detection results of the 1D-CNN model with ALTME optimizer at T1–T3, respectively. The white boxes indicate incorrect recognitions obtained by the basic 1D-CNN model.

### 3.4. Analysis of the Stability and Applicability of the ALTME Optimizer

In this study, the stability of the optimizer was analyzed by comparing the decreases in loss value of the ALTME and Adam optimizers during the training process. As shown in Figure 16, in contrast to the parameter update process of the Adam optimizer of the traditional 1D-CNN model, the parameter update process of ALTME has an update mechanism based on the second momentum term of the optimizer, and the oil-spill detection model that uses the ALTME optimizer will accumulate and learn the long-term spectral characteristic information of different batches. As the number of training iterations increases, the second momentum term of the ALTME optimizer approaches a constant, the magnitude of correction decreases, and the learning rate stabilizes. As a result, the loss of the model decreases steadily during the training process, and the model parameters are updated more smoothly so that the model is more stable at the beginning of the iteration and the convergence stage.

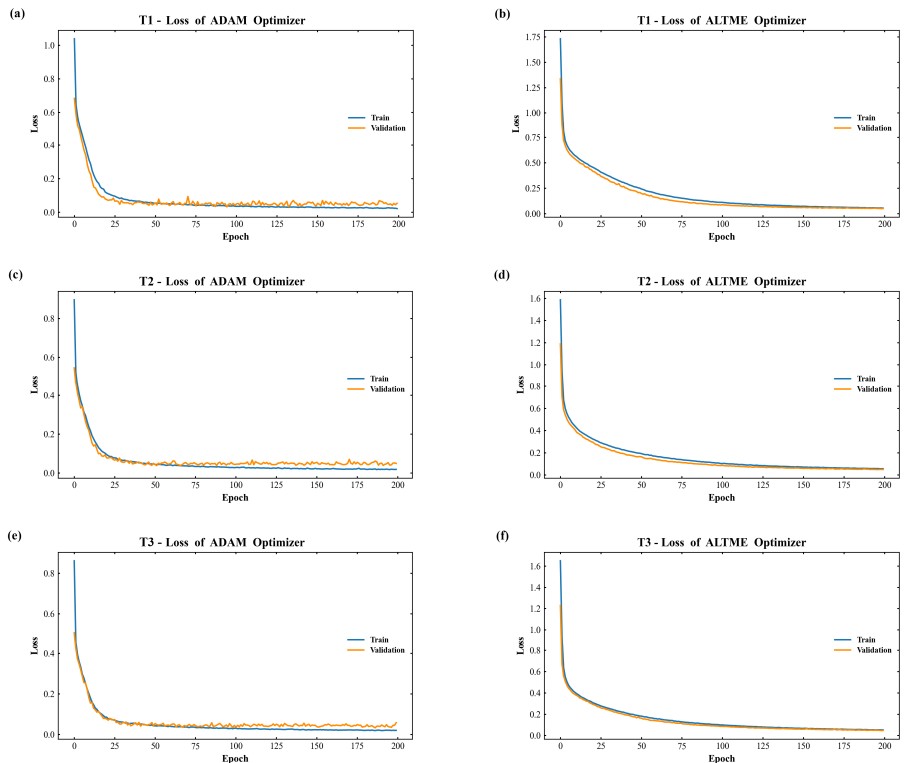

**Figure 16.** Comparison of the loss values of the ALTME and Adam optimizer. (**a**,**c**,**e**) are loss values of the Adam optimizer at T1–T3, respectively. (**b**,**d**,**f**) are loss values of the ALTME optimizer at T1–T3, respectively.

Because of the mechanism of cumulatively learning long-term spectral features in the ALTME optimizer, the amount of calculation may increase and consequently reduce the detection efficiency of the model. In this study, the effect of the ALTME optimizer on detection efficiency was analyzed, and the results are shown in Table 9. The results reveal that the detection time of the model equipped with the ALTME optimizer instead of Adam increases to varying degrees at T1–T3.

**Table 9.** Runtime of the ALTME optimizer cumulative learning weight adaptive selection experiment.

| | | Time (min) | | | | |
|---|---|---|---|---|---|---|
| **Optimizer** | **ADAM** | **ALTME ($\lambda$)** | | | | |
| **Moment** | | $1 \times 10^{-4}$ | $2 \times 10^{-4}$ | $3 \times 10^{-4}$ | $4 \times 10^{-4}$ | $5 \times 10^{-4}$ |
| T1 | 11.68 | 11.97 | 12.09 | 12.19 | 12.29 | 12.41 |
| T2 | 11.77 | 12.01 | 12.12 | 12.22 | 12.35 | 12.43 |
| T3 | 11.63 | 11.93 | 12.07 | 12.18 | 12.27 | 12.39 |

As shown in Figure 17, during the adaptive cumulative weight selection experiment, as the adaptive weight $\lambda$ gradually increases from $1 \times 10^{-4}$ to $5 \times 10^{-4}$, the detection time of the model tends to increase. Under the premise of model convergence, as $\lambda$ increases, the amount of oil-spill spectral information learned by the ALTME optimizer increases, which increases the memory occupied by the model in the graphics processing unit, thereby slowing down the overall convergence process. Because the magnitude of $\lambda$ in the experiment is only $1 \times 10^{-4}$, the update mechanism of the second momentum term of the ALTME optimizer only affects the detection efficiency of the model to a certain extent. The increase in detection time is less than 1 min, which will not incur too much computational burden in the model detection process.

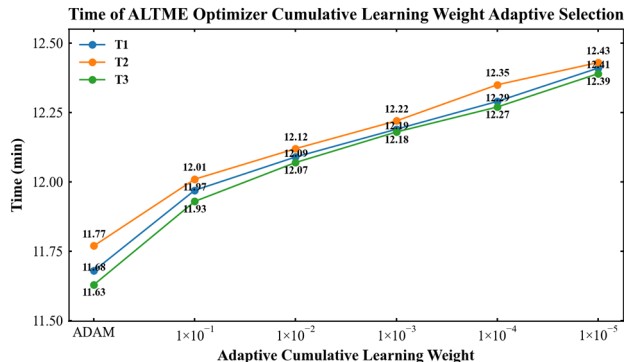

**Figure 17.** Time of ALTME optimizer cumulative learning weight adaptive selection.

Our experimental data are $1000 \times 1000 \times 125$ pixels in size with a spectral range of 450–950 nm and include nine oil-spill experiment groups. The size of the Pavia University public hyperspectral dataset is $610 \times 340 \times 103$ pixels with a spectral range of 430–860 nm and includes nine types of targets. These specifications are similar to those of the hyperspectral data obtained in our experiment. Therefore, the Pavia University public dataset was selected to verify the applicability of the proposed ALTME optimizer. The number of calibration samples in the Pavia University data is listed in Table 10. We set the ratio of training samples to test samples to 1:9, which is consistent with the ratio in the oil-spill detection experiment.

**Table 10.** Calibrated sample labels for the Pavia University data.

| Group | Class | Samples | Colors |
|:---:|:---:|:---:|:---:|
| 1# | Asphalt | 6631 | |
| 2# | Meadows | 18,649 | |
| 3# | Gravel | 2099 | |
| 4# | Trees | 3064 | |
| 5# | Painted metal sheets | 1345 | |
| 6# | Bare Soil | 5029 | |
| 7# | Bitumen | 1330 | |
| 8# | Self-Blocking Bricks | 3682 | |
| 9# | Shadows | 947 | |

As shown in Table 11, the adaptive weight $\lambda$ value of the ALTME optimizer was self-selected to be $2 \times 10^{-4}$, which enables the model to cumulatively learn multiple batches of long-term historical spectral feature information to identify the type of targets in the test area. The OA of the 1D-CNN model equipped with the ALTME optimizer is 95.91%, and the Kappa coefficient is 0.946, which are both higher than the results of the SVM and traditional 1D-CNN models. Except for the painted metal sheets (group 5), the detection model with the ALTME optimizer has the best recognition accuracy for the remaining eight types of targets.

As shown in Figure 18, the recognition results of gravel (group 3) and bitumen (group 7) by SVM and traditional 1D-CNN model are obviously misclassified, and this is even more obvious for bare soil (group 6). The 1D-CNN model with the ALTME optimizer can fully learn the spectral feature information based on the cumulative learning mechanism, effectively improving the recognition performance of various targets in the test area, and the recognition results are more consistent. The airborne hyperspectral remote sensing data used in our oil-spill detection experiment are close to the Pavia University data in terms of data specifications. From this, we conclude that the ALTME optimizer proposed in this paper can improve recognition accuracy by accumulating and learning long-term spectral information, which is suitable for the detection and analysis of hyperspectral remote sensing data. Moreover, it has applicability and generalization ability.

**Table 11.** Recognition accuracy results for the Pavia University data.

| | Index | | | | | | | | |
|---|---|---|---|---|---|---|---|---|---|
| **Model** | **SVM** | | | **1D-CNN** | | | **1D-CNN-ALTME ($\lambda = 2 \times 10^{-4}$)** | | |
| **Group** | **Recall** | **Precision** | **$F_1$-Score** | **Recall** | **Precision** | **$F_1$-Score** | **Recall** | **Precision** | **$F_1$-Score** |
| 1#-Asphalt | 93.68 | 87.44 | 0.905 | 92.34 | 93.79 | 0.931 | 94.87 | 96.47 | **0.957** |
| 2#-Meadows | 99.05 | 90.50 | 0.946 | 97.28 | 93.35 | 0.953 | 98.49 | 98.28 | **0.984** |
| 3#-Gravel | 67.72 | 82.96 | 0.746 | 77.14 | 82.28 | 0.796 | 80.38 | 90.85 | **0.853** |
| 4#-Trees | 90.25 | 96.59 | 0.933 | 89.52 | 94.60 | 0.920 | 96.48 | 97.49 | **0.970** |
| 5#-Painted metal sheets | 99.67 | 99.83 | **0.998** | 99.26 | 97.72 | 0.985 | 99.55 | 99.85 | 0.997 |
| 6#-Bare Soil | 68.39 | 96.33 | 0.800 | 79.94 | 90.32 | 0.848 | 95.11 | 95.22 | **0.952** |
| 7#-Bitumen | 48.29 | 89.20 | 0.627 | 87.05 | 86.54 | 0.868 | 91.88 | 89.99 | **0.909** |
| 8#-Self-Blocking Bricks | 89.92 | 80.65 | 0.850 | 88.44 | 84.78 | 0.866 | 93.32 | 85.69 | **0.893** |
| 9#-Shadows | 99.88 | 100.00 | **0.999** | 99.88 | 99.65 | 0.998 | 99.79 | 100.00 | **0.999** |
| OA (%) | | 90.12 | | | 91.97 | | | **95.91** | |
| Kappa | | 0.866 | | | 0.893 | | | **0.946** | |
| Time (min) | | 4.46 | | | **2.21** | | | 2.63 | |

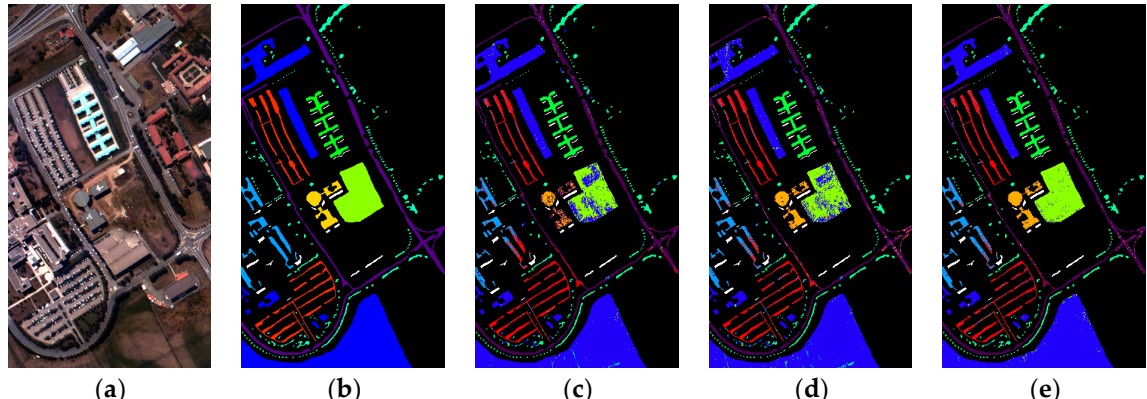

(**a**)      (**b**)      (**c**)      (**d**)      (**e**)

**Figure 18.** Recognition results for the public hyperspectral remote sensing data of Pavia University. (**a**) True color image (red: 50 band, green: 23 band, and blue: 5 band). (**b**) Sample labels. (**c**) Results of SVM. (**d**) Results of the 1D-CNN. (**e**) Results of 1D-CNN with the ALTME optimizer.

### 3.5. Feasibility Analysis of Oil-Spill Detection Based on Thermal Infrared Remote Sensing

In this study, a feasibility analysis of marine oil-spill detection based on UAV thermal infrared remote sensing technology was also performed. Using the thermal infrared sensor carried by the DJI-Mavic2 drone, a thermal infrared image of oil spills (uncalibrated thermal radiation brightness data) was obtained at 11:20 h. The wavelength range of the sensor is 8–14 μm, and the resolution is 480 × 640 pixels. The low spatial resolution of the thermal infrared image makes it difficult to carry out global sample calibration, so we randomly selected pure samples to conduct the oil-spill thermal infrared detection experiment and verification. Because of the limited endurance of the UAV, only one effective thermal infrared oil-spill image was obtained. The image and sample labels are shown in Figure 19.

The Berrcom JXB-178 (Berrcom, China) infrared thermometer was used to obtain the surface temperature (kinetic temperature) of the oil and seawater at T1–T3, as shown in Table 12. The thermal infrared oil-spill image was collected at 11:20 h, and the surface temperature at T1 was measured at 11:30 h, which is a difference of only 10 min. Therefore, we used the surface temperature data at T1 to carry out the correlation analysis with the thermal radiation brightness.

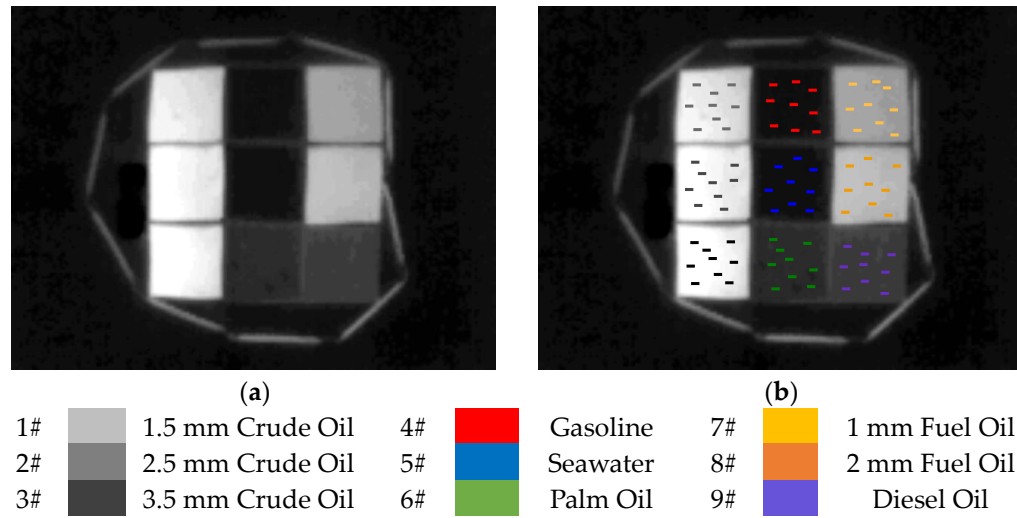

| 1# | 1.5 mm Crude Oil | 4# | Gasoline | 7# | 1 mm Fuel Oil |
| 2# | 2.5 mm Crude Oil | 5# | Seawater | 8# | 2 mm Fuel Oil |
| 3# | 3.5 mm Crude Oil | 6# | Palm Oil | 9# | Diesel Oil |

**Figure 19.** (**a**,**b**) are thermal infrared remote sensing image and data labels.

**Table 12.** Temperature information of experimental oil and seawater.

| | Temperature (°C) | | |
|---|---|---|---|
| **Group** | **T1** | **T2** | **T3** |
| 1#-1.5 mm Crude | 44.5 | 41.6 | 30.7 |
| 2#-2.5 mm Crude | 45.2 | 42.5 | 30.9 |
| 3#-3.5 mm Crude | 47.5 | 43.1 | 30.9 |
| 4#-Gasoline | 24.8 | 24.7 | 23.4 |
| 5#-Seawater | 25.3 | 24.9 | 24.5 |
| 6#-Palm Oil | 28.9 | 28.1 | 26.0 |
| 7#-1 mm Fuel Oil | 37.2 | 36.8 | 29.4 |
| 8#-2 mm Fuel Oil | 40.3 | 39.5 | 29.9 |
| 9#-Diesel Oil | 28.7 | 26.5 | 25.9 |

As shown in Figure 20, the surface temperature of each experimental group at T1 is positively correlated with the thermal radiation brightness with an $R^2$ of 0.994. We fit the kinetic temperature calibration polynomial model and performed batch thermal infrared temperature calibration on random samples of the oil spills. The thermal infrared temperature data of oil spills at T1 after calibration is shown in Table 13.

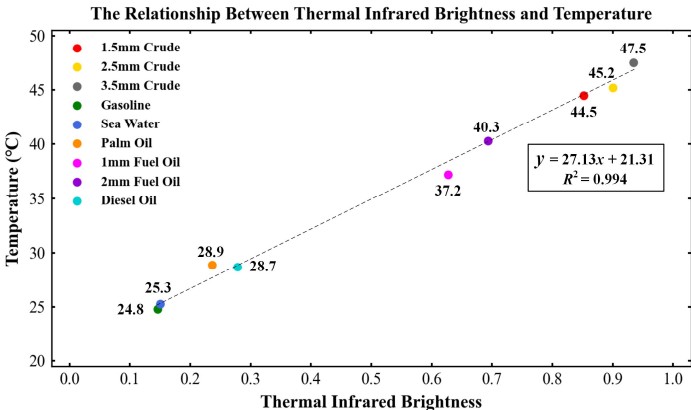

**Figure 20.** Relationship between thermal infrared brightness and oil spill temperature.

**Table 13.** Thermal infrared temperature data of oil spill at T1.

| Group | Temperature Information (°C) | | | |
|---|---|---|---|---|
| | Average | Max | Min | STD |
| 1#-1.5 mm Crude | 44.4 | 46.0 | 42.3 | 0.97 |
| 2#-2.5 mm Crude | 45.7 | 47.2 | 44.6 | 0.61 |
| 3#-3.5 mm Crude | 46.6 | 48.1 | 43.1 | 1.06 |
| 4#-Gasoline | 25.3 | 25.5 | 24.8 | 0.13 |
| 5#-Seawater | 25.4 | 25.7 | 25.2 | 0.06 |
| 6#-Palm Oil | 27.7 | 30.1 | 26.7 | 0.83 |
| 7#-1 mm Fuel Oil | 38.3 | 40.9 | 37.4 | 0.79 |
| 8#-2 mm Fuel Oil | 40.1 | 41.2 | 36.1 | 1.19 |
| 9#-Diesel Oil | 28.8 | 29.1 | 27.3 | 0.48 |

As the results in Table 13 show, the oil film thermal infrared temperature of the heavy-oil experimental group is positively correlated with the thickness. A thicker oil film is associated with a higher temperature. In addition, the standard temperature deviations (STDs) of the heavy-oil groups are larger than that of the seawater and light-oil groups. We believe that the temperature STD can be used to characterize the uniformity of the oil film distribution on the sea surface. The light-oil films tend to diffuse evenly, so the temperature of each position of the oil film is relatively uniform with a small STD. Heavy oils have high surface viscosity, large surface roughness, and poor uniformity, leading to a relatively high-temperature STD.

Because of the low dimensionality of the thermal infrared remote sensing image data, deep learning is not suitable in this case. Instead, an SVM model was used to detect the oil spill, and the detection accuracy is shown in Table 14. The OA reaches 74.58%, Kappa reaches 0.714, and the detection time is 1.19 min. The detection accuracy $F_1$-score for heavy-oil films is larger than 0.634, and the recognition accuracy $F_1$-score for light-oil types is larger than 0.551. These experimental results demonstrate that there is potential for marine oil-spill detection research based on thermal infrared remote sensing images.

**Table 14.** Accuracy of thermal infrared oil spill detection based on SVM.

| Group | Index | | |
|---|---|---|---|
| | Recall (%) | Precision (%) | $F_1$-Score |
| 1#-1.5 mm Crude | 74.24 | 68.37 | 0.712 |
| 2#-2.5 mm Crude | 67.68 | 59.56 | 0.634 |
| 3#-3.5 mm Crude | 68.18 | 87.66 | 0.767 |
| 4#-Gasoline | 38.38 | 97.44 | 0.551 |
| 5#-Seawater | 98.99 | 61.64 | 0.760 |
| 6#-Palm Oil | 84.85 | 80.38 | 0.826 |
| 7#-1 mm Fuel Oil | 85.35 | 76.82 | 0.809 |
| 8#-2 mm Fuel Oil | 74.24 | 83.52 | 0.786 |
| 9#-Diesel Oil | 79.29 | 83.96 | 0.816 |
| OA (%) | | 74.58 | |
| Kappa | | 0.714 | |
| Time (min) | | 1.19 | |

The temperature difference (TD) between the oil films determines the ability of thermal infrared remote sensing to detect oil spills. As shown in Table 12, from T1 to T3, as time passes, the oil film surface temperature of each experimental group tends to decrease. The gradual decrease in the solar elevation angle over time led to the decrease in sunlight intensity and the reduction in the heat absorbed by the oil film surface so that the surface temperature of the oil spill decreased over time. Heavy oil has a strong absorbing effect on sunlight, which increases the surface temperature of the oil film. Therefore, the surface temperature of the oil film of the heavy-oil group is higher than that of the light-oil and

seawater groups at T1–T3. Moreover, for the heavy-oil groups, the surface temperature increased with increases in the thickness of the oil film. The surface temperatures of the crude oil and fuel oil groups differ substantially: the overall surface temperature of the crude oil groups is higher than that of the fuel oil groups.

At T3, the TDs between the light-oil and seawater groups was less than $\pm 1.6\ °C$, the TD between the crude oil groups was less than $\pm 0.2\ °C$, and the TD of the fuel oil group was less than $\pm 0.5\ °C$. From this, we can infer that under strong light intensity, there is an obvious TD between light oils and seawater as well as between heavy-oil films with different thicknesses. The detection results further show that thermal infrared remote sensing technology has a certain potential for oil-spill detection. However, under low light intensity conditions, the TDs are small for light oils and seawater as well as for different heavy-oil films. This will affect the sensitivity of thermal infrared detection, thereby affecting the accuracy of oil-spill detection. In the future, hyperspectral and thermal infrared remote sensing technology will be combined to achieve coaxial multi-dimensional data acquisition of marine oil spills, which is expected to increase the feature extraction space of the deep learning model and improve the accuracy of oil-spill detection based on the ALTME optimizer proposed in this paper.

## 4. Conclusions

### 4.1. Conclusions

Marine oil-spill accidents seriously threaten both the marine ecological environment and human health. The type of oil spills and the thickness of the oil films are important basic items of information needed for scientific decision-making at the oil-spill site. In this study, an experimental setup was constructed to simulate real marine oil-spill scenarios to the greatest extent, and accuracy verification for the detection model was carried out. UAV imaging hyperspectral remote sensing technology was then used to investigate rapid and effective oil-spill detection. The ALTME optimizer, which is suitable for the characteristics of hyperspectral imagery and can adapt to different time phases of oil-spill scenarios, was proposed. Experimental results show that the ALTME optimizer can assign adaptive cumulative learning weights to multiple batches of long-term oil-spill spectral information to fully learn and utilize the spectral information. It effectively overcomes the problem of poor spectral separability among light oils, deeply learns the spectral differences between thick oil films, and then accurately identifies the type of oil spills and detects the thickness of thick oil films.

In this study, the ALTME optimizer was also evaluated on the Pavia University hyperspectral data, which are similar to the experimental data in terms of specifications. The results show that the optimizer exhibits excellent generalization and applicability, which make it suitable for hyperspectral detection research. This study also actively explored oil-spill detection based on thermal infrared remote sensing. Under strong sunlight, thermal infrared remote sensing technology has oil-spill detection potential.

### 4.2. Perspectives and Future Works

In this study, the sunlight intensity was obtained by calculating the solar elevation angle and then used to analyze its influence on the oil-spill spectrum curves. In the future, we plan to use a photometer to directly measure the sunlight intensity at different times. This study is an exploratory study of the oil-spill detection ability of the UAV hyperspectral sensor under ideal conditions. We plan to carry out oil-spill detection research under different sunlight conditions, strong winds, and other rough conditions in the future so as to explore the detection ability of the model proposed in this paper. In this study, we found that thick oil films with different thicknesses can be qualitatively detected by UAV hyperspectral sensors. In future research, we will carry out research on the quantitative inversion of oil film thickness on this basis.

Moreover, we found that thermal infrared remote sensing technology showed a certain potential for oil-spill detection under strong sunlight conditions. Our team intends to

develop a coaxial multi-dimensional integrated sensor for both hyperspectral and thermal infrared data to obtain multi-dimensional remote sensing data of oil spills on the sea surface, thereby increasing the feature space of the deep learning model and improving the detection accuracy of marine oil spills.

**Author Contributions:** Conceptualization, J.Z. and Y.M.; methodology, J.Z. and Y.M.; software, Z.J.; validation, Z.J. and X.M.; formal analysis, Z.J. and X.M.; investigation, Z.J. and X.M.; resources, J.Z. and Y.M.; data curation, Y.M. and Z.J.; writing—original draft preparation, Z.J. and Y.M.; writing—review and editing, Z.J. and Y.M.; visualization, Z.J. and Y.M.; supervision, J.Z., Y.M. and M.X; project administration, J.Z. and Y.M.; funding acquisition, J.Z. and Y.M. All authors have read and agreed to the published version of the manuscript.

**Funding:** This research was funded by "The National Natural Science Foundation of China NO. 61890964" and "The National Natural Science Foundation of China NO. U1906217".

**Acknowledgments:** The authors would like to thank the Editor-in-Chief, the Associate Editor, and the reviewers for their insightful comments and suggestions. We thank Kimberly Moravec, from Liwen Bianji (Edanz), for editing the English text of a draft of this manuscript.

**Conflicts of Interest:** The authors declare no conflict of interest.

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
