# Peer review of "Hyperspectral Remote Sensing Detection of Marine Oil Spills Using an Adaptive Long-Term Moment Estimation Optimizer"

_remotesensing, doi:10.3390/rs14010157_

Round 1
Reviewer 1 Report
The manuscript is interesting and would be beneficial to the potential readers. The methods described in the manuscript addressed several key main questions during immediate spill responses such as rapid identification of oil types and their thickness. The manuscript also addressed several currently available techniques based on sensor approaches and are up-to-date with current technologies. Apart from that, as it has been a limitation to most other studies, this study compared the technique accuracy based on a wide range of oil type and thickness based on several models.
However, I would suggest that another main factor which the authors should consider and describe is the ability of their method to handle rough weather conditions such as strong winds with waves and under low or high sunlight conditions. These factors have been an issue for most other remote sensing studies. Apart from that, balancing of camera during imaging is also another important factor where a slight change in angle would fluctuate the entire image accuracy.
Author Response
Response to dear reviewer :
Thank you for your valuable comments, we think your comments will be very helpful to our manuscript. Your opinion makes our paper more rigorous and scientific.
- What you said is really correct. Environmental factors are indeed an important factor affecting the observation of oil spills on the sea today, especially the influence of the sun glint. In this experiment, we took the flying method of avoiding the sun glint to avoid the influence of the sun glint. This method is a common method for detecting oil spills on the sea surface in the world, such as the USGS and Hu’s And we have made a supplementary explanation in the manuscript according to your suggestion.
- In addition, what you said is really correct,the strong winds and waves interfere very strongly with oil spills on the sea surface. Our research is an exploratory research to explore the ability of the airborne hyperspectral sensor to detect oil spills on the sea surface in an ideal environment. We have purchased wave making machines, and we plan to carry out sea-surface oil spill detection experiments in rough environments this year. This part of the explanation has been supplemented in the “Perspectives & futur works” section of the paper.
- You are correct. The balance of the camera is another important factor. We chose the DJI M600Pro UAV as the sensor carrying platform. The M600Pro is equipped with A3Pro flight control system and Zenmuse gimbal, which has good stabilityand balance ability. And this was also our best choice for purchasing at that time. And we have made a supplementary explanation in the manuscript according to your suggestion.
Thank you for your valuable comments. We have revised the manuscript carefully and in detail according to your advice.

Reviewer 2 Report
This manuscript proposes an experimental protocol to classify oil spills using hyperspectral remote sensing data. Nine different classes were tested in this experimental protocol, including seawater, two groups of heavy oil, and three groups of light oil of different thicknesses. Images were acquired at three different times. Three models were tested (SVM, GRU and 1D_CNN) for the purpose of classification. The 1D_CNN showed the best performance and was optimized by incorporating the temporal component using the Adam and ALTME moment estimation optimizers. As a final step, the authors showed the potential of using thermal infrared brightness to estimate the temperature of oil spills. The topic of the paper is very interesting. The results are promising and very accurate. However, I have one criticism regarding the methodological approach is that the classification evaluation is done using the calibration data. The weights of the deep learning and machine learning classifiers used are optimized for this data, the reason I think the authors achieved such high classification accuracy. However, some very interesting ideas are presented in the paper. In general, the sections of the manuscript are clear, and it can be considered for publication with some minor corrections. My comments and suggestions are included in the pdf file.

Author Response
Response to dear reviewer :
Thank you for your valuable comments, we think your comments will be very helpful to our manuscript. Your opinion makes our paper more rigorous and scientific.
- they are already included in the above 5 types!
Thank you for your advice. We have revised it in our manuscript according to your advice.
- This paragraph is part of the methodology section.
Thank you for your advice. We want to highlight the innovations and contributions of our research in the introduction. And we have revised it in methodology section, and streamlined this part of the content according to your advice.
- a section of the images preprocessing is lacking ( radiometric and reflectance computation)
Thank you for your advice. The S185 hyperspectral sensor obtains radiance data. Based on the Python and Cubert-Touch platforms, we use AZ-WS20 standard plate to convert radiance into remote sensing reflectance. We have supplemented this part of the content in the manuscript according to your suggestions.
- you need to add a more global map to thesetwo zoomed-in figures.
Thank you for your advice. We have added a more global map to these two zoomed-in figures according to your suggestions.
- Please define these accronyms.
Thank you for your advice. We have revised it in our manuscript according to your advice. We changed “PVC” to “ PVC (Polyvinyl Chloride)”.
- I got confused identifying the oils from left to right. You can label them by the number of small boxes.
Thank you for your advice. We have modified the figures and marked the corresponding oil labels on the figures according to your advice.
- You need to specify which groups are T1, T2 and T3 on the title of the figure. For example: T1 (a and d) and so on.
Thank you for your advice. We have revised it in our manuscript according to your advice.
- Results and discussion
Thank you for your advice. We have changed “Results and Analysis” to “Results and discussion
” according to your advice.
- Conclusions
Thank you for your advice. We have changed “Conclusion and Discussion” to “Conclusions” according to your advice.
- Perspectives & futur works
Thank you for your advice. We have changed “Discussion” to “Perspectives & futur works” according to your advice.
Thank you for your valuable comments. We have revised the manuscript carefully and in detail according to your advice.

Reviewer 3 Report
Minor Revision.
Thanks to the authors for the exhaustive study.
Only some minor comments: check in general in all the text the way to mention References, please refer to guide line.
LINE 218: correct wolud with would
Author Response
Response to dear reviewer :
Thank you for your valuable comments, we think your comments will be very helpful to our manuscript. Your opinion makes our paper more rigorous and scientific.
We have rechecked the full text and revised the format of the manuscript and references carefully.
Thank you for your valuable comments.
